# Cavin-3 dictates the balance between ERK and Akt signaling

Victor J Hernandez[1], Jian Weng[1], Peter Ly[1], Shanica Pompey[1], Hongyun Dong[1], Lopa Mishra[2], Margaret Schwarz[3], Richard GW Anderson[1†], Peter Michaely[1]*

[1]Department of Cell Biology, University of Texas Southwestern Medical Center, Dallas, United States; [2]Department of Gastroenterology, Hepatology, and Nutrition, University of Texas MD Anderson Cancer Center, Houston, United States; [3]Department of Pediatrics, University of Texas Southwestern Medical Center, Dallas, United States

**Abstract** Cavin-3 is a tumor suppressor protein of unknown function. Using both in vivo and in vitro approaches, we show that cavin-3 dictates the balance between ERK and Akt signaling. Loss of cavin-3 increases Akt signaling at the expense of ERK, while gain of cavin-3 increases ERK signaling at the expense Akt. Cavin-3 facilitates signal transduction to ERK by anchoring caveolae to the membrane skeleton of the plasma membrane via myosin-1c. Caveolae are lipid raft specializations that contain an ERK activation module and loss of the cavin-3 linkage reduces the abundance of caveolae, thereby separating this ERK activation module from signaling receptors. Loss of cavin-3 promotes Akt signaling through suppression of EGR1 and PTEN. The in vitro consequences of the loss of cavin-3 include induction of Warburg metabolism (aerobic glycolysis), accelerated cell proliferation, and resistance to apoptosis. The in vivo consequences of cavin-3 knockout are increased lactate production and cachexia.

*For correspondence: Peter.
Michaely@utsouthwestern.edu

†Deceased

Competing interests: The authors declare that no competing interests exist.

## Introduction

Cavin-3 (*PRKCDBP, hSRBC*) is a tumor suppressor protein of unclear function. In humans, cavin-3 is encoded in the 11p15.5 tumor suppressor locus and loss of cavin-3 expression is common in many epithelial and glial derived cancers (*Zochbauer-Muller et al., 2005*; *Lee et al., 2008*; *Martinez et al., 2009*; *Tong et al., 2010*; *Caren et al., 2011*; *Lee et al., 2011*). Cavin-3 expression is also absent in many cancer cell lines and ectopic expression of cavin-3 in these cells is sufficient to suppress their tumorigenesis in athymic mice (*Xu et al., 2001*; *Lee et al., 2011*). How cavin-3 expression suppresses tumorigenesis is not clear; however, forced over-expression of cavin-3 can induce G1 arrest and promote apoptosis (*Lee et al., 2011*), suggesting that cavin-3 may suppress mitogenic signaling.

Cavin-3 is one of four cavin family members, all of which are localized to caveolae (*Bastiani et al., 2009*). Caveolae are invaginated, lipid-raft microdomains of the plasma membrane that may play roles in mitogenic signaling because a population of EGF, PDGF, and insulin receptors have been visualized by immuno-EM either in or adjacent to caveolae (*Liu et al., 1996*; *Foti et al., 2007*; *Nagy et al., 2010*). Caveolae are not, however, required for cell proliferation because caveolae are not present in all cell types, caveolae do not appear until late in embryogenesis and animals with mutations that prevent caveolae formation are viable and of normal size (*Engelman et al., 1998*; *Drab et al., 2001*; *Razani et al., 2001, 2002*; *Fang et al., 2006*; *Liu et al., 2008*). Caveolae may instead limit cell proliferation because some mutations that prevent caveolae formation are associated with hyperplasia and fibrosis in lung (*Drab et al., 2001*; *Razani et al., 2001*).

Cavins likely serve both structural and functional roles in caveolae through their interactions with the caveolin family of integral membrane proteins. Cavin-1 provides a structural function by binding to

**eLife digest** The plasma membrane separates cells from their environment, and surface receptors in this membrane allow cells to respond to changes in their environment by converting external cues into intracellular signals. This process, which is known as signal transduction, plays a central role in the biology of cells, and abnormal signaling is a common cause of human disease. In cancer for example, signals tend to be too strong or they are sent at the wrong time.

Signal transduction frequently occurs at specialized regions of the plasma membrane. Caveolae are small indentations of the plasma membrane that comprise one type of signaling specialization. A protein that is concentrated in caveolae, cavin-3, suppresses tumor formation and is commonly absent from cancer cells. These observations suggest that cavin-3 participates in signal transduction and pathways that are associated with cancer, but the details of this involvement are not well understood.

Hernandez et al. now show that cavin-3 controls the balance between two key intracellular signals, ERK and Akt. High levels of cavin-3 promote activation of the ERK signaling pathway but suppress activation of the Akt signaling pathway. Loss of cavin-3 has the opposite effect, activating Akt at the expense of ERK. The consequences of loss of cavin-3 include accelerated cell proliferation, the induction of Warburg metabolism (a metabolic state that supports rapid cell division), and the suppression of the apoptosis pathway. (Suppression of this pathway, which leads to cell death, allows cancer cells to proliferate in the body.) While deletion of the cavin-3 gene in mice is not sufficient to cause spontaneous cancer, animals that are deficient in cavin-3 die prematurely of cachexia, a tissue wasting sequela experienced by nearly half of all cancer patients.

Hernandez et al. also show that cavin-3 influences cellular signaling by linking caveolae to the membrane skeleton—a network of proteins that underlies the plasma membrane. This linkage is necessary to ensure that cells have the correct abundance of caveolae, and it also facilitates signal transduction to the ERK signaling pathway. ERK activation in this context drives expression of two proteins, EGR1 and PTEN, which suppress Akt signaling. Hernandez et al. propose that the membrane skeleton functions as a scaffold that adaptors, such as cavin-3, use to assemble signaling modules with surface receptors for the purpose of controlling the signal transduction output from these receptors.

the principal caveolin, caveolin-1, stabilizing the invaginated morphology of caveolae and providing an interaction surface for other cavins (*Hill et al., 2008*; *Liu and Pilch, 2008*; *Bastiani et al., 2009*). Cavin-2 and Cavin-4 show restricted expression and may serve cell-type specific functions (*Bastiani et al., 2009*; *Hansen et al., 2013*). By contrast, cavin-3 is broadly expressed and has been proposed to function in caveolae internalization (*McMahon et al., 2009*). How cavin-3 might participate in cellular signaling is not clear.

Here we show that cavin-3 dictates the balance between ERK and Akt signaling with consequences for cell metabolism, apoptosis and cell proliferation. We also characterize the molecular mechanisms by which cavin-3 influences cellular signaling.

## Results

### Loss of cavin-3 alters cell signaling, induces Warburg metabolism and inhibits apoptosis

We employed two model systems to investigate whether loss of cavin-3 influences cell signaling. The first model system examined acute effects by depleting cavin-3 with siRNA in human SV589 fibroblasts over a time course of 15 days (*Figures 1–3*). The second model system examined chronic effects by knocking out the cavin-3 locus in the mouse and characterizing the consequences of loss of cavin-3 in embryonic fibroblasts (*Figure 4*). Because loss of cavin-3 is associated with cancer in multiple tissues and cancer is associated with elevated mitogenic signaling, we hypothesized that loss of cavin-3 might augment signaling in response to growth factors.

Contrary to expectation, microarrays showed that knockdown of cavin-3 suppressed the ability of EGF to induce most immediate early (IE) response genes (*Figure 1A*). Suppressed transcripts fell into

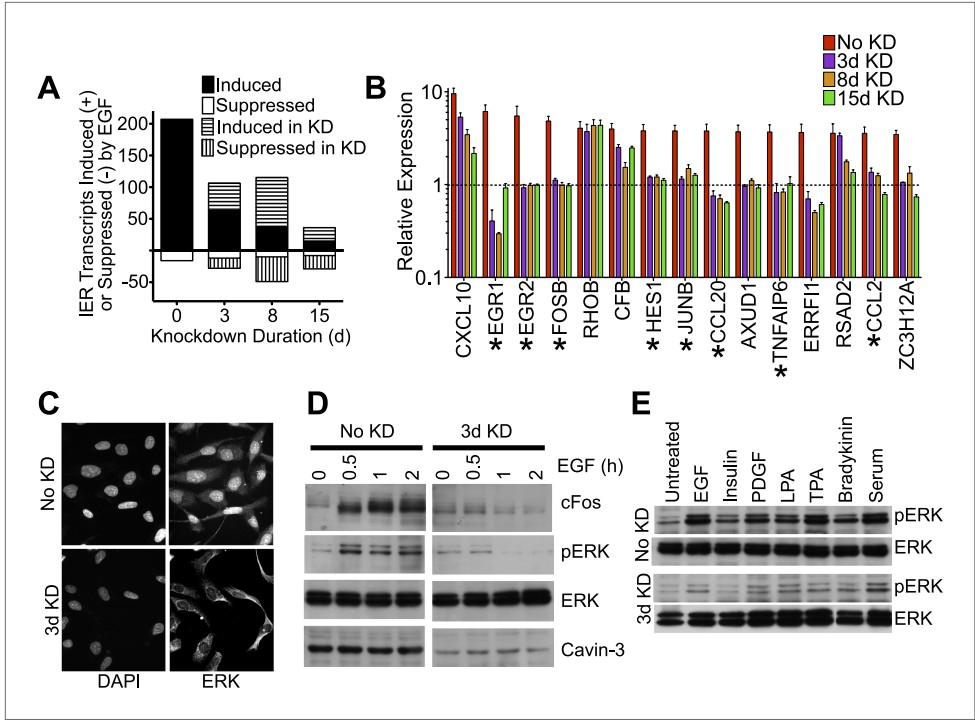

**Figure 1**. Knockdown of cavin-3 suppresses mitogen-dependent ERK activation. (**A**) Loss of cavin-3 suppresses IE response. Human SV589 fibroblasts were mock treated (0 day) or treated with cavin-3 siRNA for 2, 7 or 14 days. Knockdown was maintained by splitting and re-transfecting cells with cavin-3 siRNA on days 5, 9, and 12. Cells were then serum starved for 20 hr and RNA from cells was either harvested immediately or following treatment with 100 ng/ml EGF for 1 or 3 hr. Bars indicate the number of transcripts whose mean expression increased or decreased twofold following 1 hr, but not 3 hr, treatment. Solid and open bars indicate the number of transcripts common to transcripts induced or suppressed in the absence of knockdown. Hashed bars indicate knockdown-specific transcripts. Microarrays were performed in triplicate for 47,323 transcripts. Complete microarray data is provided (Dryad: *Michaely et al., 2013*). (**B**) Leading edge analysis shows that loss of cavin-3 impairs ERK activation by EGF. Transcripts induced by 1 hr but not 3 hr of EGF stimulation were ordered based upon fold-induction using microarray data collected from cells without knockdown (No KD). Fold-inductions for the top 15 transcripts are shown together with fold-inductions for the same transcripts in the 3-day, 8-day and 15-day knockdowns. All data are means ± SD, n = 3. (*) indicates genes for which published data has identified transcriptional regulation by ERK (*Agarwal et al., 1995*; *Gille et al., 1995*; *Cohen, 1996*; *Ochsner et al., 2003*; *Lin et al., 2004*; *Hosokawa et al., 2005*; *Stockhausen et al., 2005*; *Bradley et al., 2008*). (**C**) Cavin-3 knockdown impairs ERK translocation to the nucleus. Fibroblasts were mock treated (No KD) or treated with cavin-3 siRNA for 2 days, starved of serum for 20 hr, stimulated with EGF for 15 min, fixed and immunostained for total ERK. Nuclei were stained using DAPI. (**D**) Knockdown impairs cFos induction. Fibroblasts were mock treated (No KD) or treated with cavin-3 siRNA for 2 days, starved 20 hr for serum and stimulated with EGF for the indicated time. Cells were then lysed and immunoblotted for the indicated protein. (**E**) Loss of cavin-3 suppresses ERK activation by diverse mitogens. Fibroblasts were mock treated (No KD) or treated with cavin-3 siRNA, cultured for 2 days in serum, serum starved overnight and stimulated with the indicated mitogen. Cell lysates were immunoblotted for indicated proteins.

two categories: those that were fully suppressed by 3 days of knockdown and those that fell gradually during the 15-day time course. Many IE transcripts in the first group encoded proteins whose expression is driven by ERK (*Figure 1A,B*). This observation suggested that loss of cavin-3 suppressed ERK signaling and 3 days of knockdown proved sufficient to inhibit EGF-dependent phosphorylation of ERK (pERK), translocation of ERK to the nucleus and induction of the ERK-responsive transcription factor, cFos (*Figure 1C,D*). The impact of cavin-3 depletion on ERK signaling was not specific to EGF because 3-day knockdown also impaired ERK activation by serum and diverse stimuli (*Figure 1E*). Cavin-3 expression is thus necessary for efficient ERK activation.

pERK is a potent driver of cell proliferation and knockdown of cavin-3 initially slowed cell growth; however, proliferation returned to a normal rate after 3 days and exceeded the normal rate after

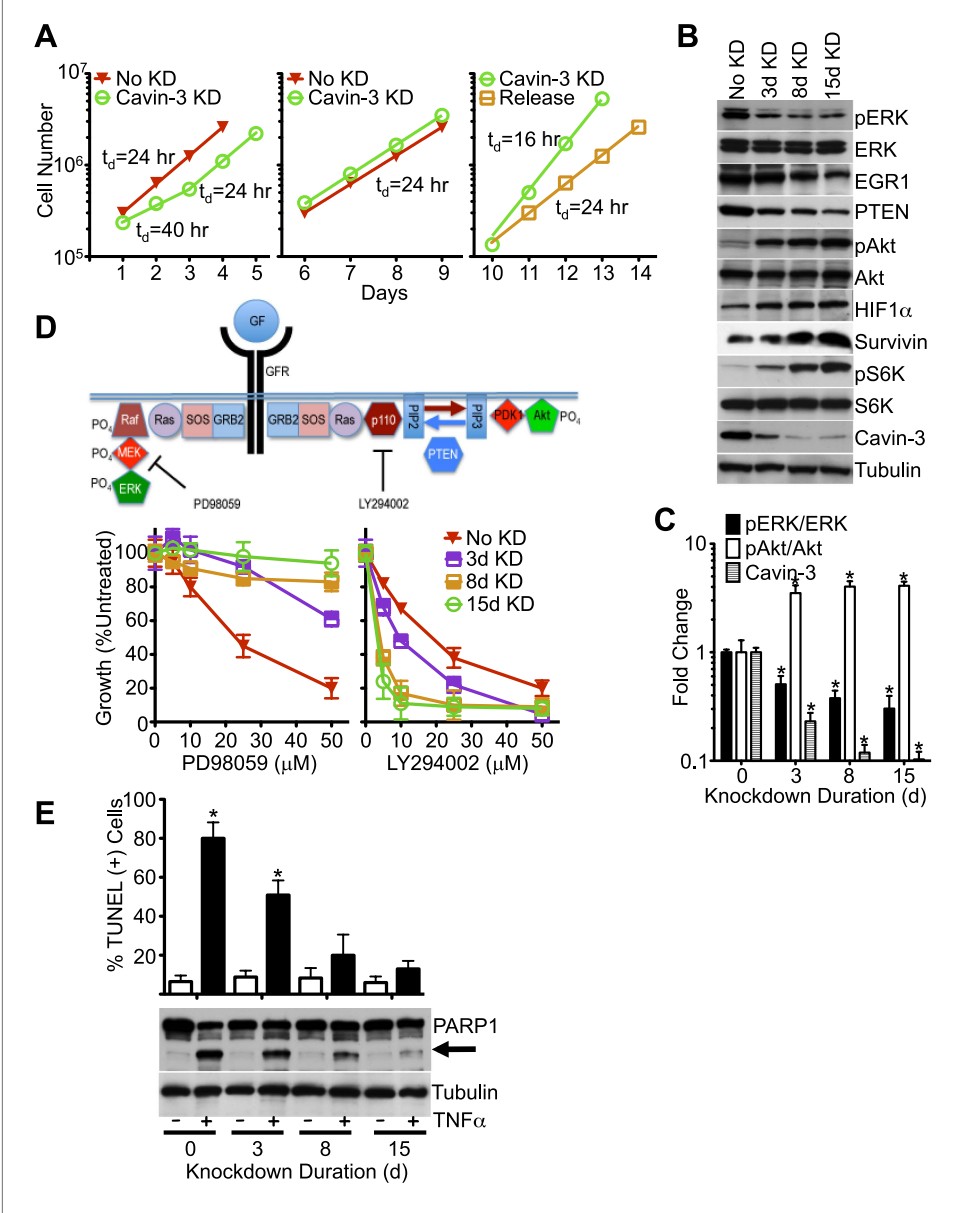

**Figure 2**. Knockdown of cavin-3 activates Akt. (**A**) Loss of cavin-3 first slows then accelerates cell proliferation. Fibroblasts were mock treated or treated with siRNA against cavin-3 and counted daily. Cells were re-treated and re-plated on day 5. On day 9, cells that had been treated with cavin-3 siRNA were either re-treated or allowed to recover from cavin-3 depletion (Release). Doubling times (td) are indicated. (**B**) Protein profile of cavin-3 knockdown cells over time. SV589 fibroblasts were mock treated (No KD) or treated with cavin-3 siRNA. Cells were split and re-treated with siRNA on days 5, 9, and 12. Cell lysates were prepared when indicated and immunoblotted for the indicated proteins. (**C**) Knockdown augments pAkt at the expense of pERK. pERK, ERK, pAkt, Akt and cavin-3 immunoblot staining was quantified by densitometry. Data were normalized to No KD (0 day) and are means ± SEM, n = 3. *p<0.05 as compared to No KD. (**D**) Knockdown confers resistance to PD98059 and sensitivity to LY294002. PD98059 interrupts the signaling pathway from growth factor receptors (GFRs) to ERK by inhibiting MEK. LY294002 interrupts the signaling pathway from GFRs to Akt by inhibiting the p110 subunit of PI3K. Cells were depleted of cavin-3 for the indicated number of days and treated with the indicated concentrations of PD98059 or LY294002 for 24 hr. Data are shown as a percentage of untreated and are means ± SEM, n = 6. (**E**) Knockdown suppresses TNFα-dependent apoptosis. The indicated cells were treated overnight with 10 µg/ml cyclohexamide alone (−) or in combination with 10 ng/ml TNFα (+) and assayed for apoptotic cells by TUNEL staining (top) and PARP1 cleavage (arrow, bottom). TUNEL data are means ± SEM, from three independent experiments. *p<0.05 as compared to no TNFα.

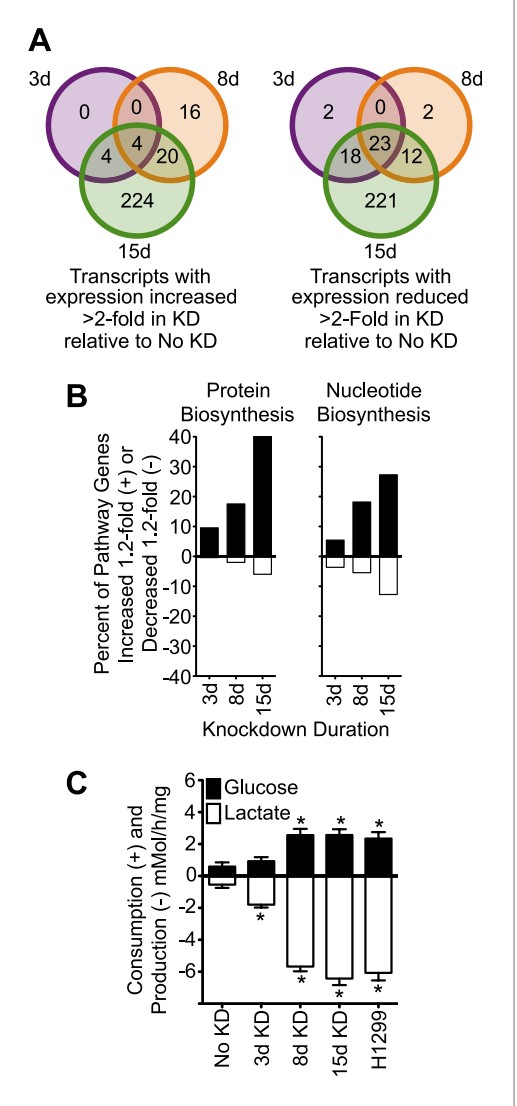

**Figure 3**. Knockdown of cavin-3 increases transcription of biosynthetic genes and induces aerobic glycolysis. (**A**) Knockdown of cavin-3 progressively alters gene expression. RNA from SV589 fibroblasts grown in normal medium was harvested from mock-transfected cells or cells treated with cavin-3 siRNA for 3, 8, or 15 days. Transcripts that were increased or decreased twofold in triplicate microarrays are presented as Venn diagrams. Numbers indicate the number of transcripts common or unique to each knockdown. Complete microarray data is available at Dryad (**Michaely et al., 2013**). (**B**) Prolonged knockdown augments many protein and nucleic acid biosynthetic components. The percent of gene transcripts with either >20% increase or >20% decrease over no knockdown in heat maps for protein and nucleic acid synthesis are plotted. Heat maps are provided in **Figure 3—figure supplement 1**. (**C**) Knockdown increases fermentative glycolysis. Glucose consumption and lactate production were measured over 8 hr by colorimetric assay. Data are
*Figure 3. Continued on next page*

10 days of knockdown (**Figure 2A**). pERK levels did not recover during the time course (**Figure 2B**), indicating that cavin-3 depleted cells compensated through other means. To identify the mechanism, we queried the IE microarray data and noted that the transcript whose expression showed the greatest suppression following cavin-3 depletion was early growth response protein 1 (EGR1) (**Figure 1B**). EGR1 is an ERK-induced transcription factor that drives expression of phosphatase and tensin homolog protein (PTEN) (**Yu et al., 2011**). PTEN opposes the action of phosphatidylinositol 3-kinase (PI3K) by dephosphorylating phosphatidylinositol 3,4,5-trisphosphate (PIP3) back to phosphatidylinositol 4,5-bisphosphate (PIP2). PIP3 generation by PI3K is necessary for the recruitment and activation of Akt (**Engelman et al., 2006**). Knockdown of cavin-3 caused progressive loss of both EGR1 and PTEN. Loss of EGR1 and PTEN coincided with a fivefold increase in steady state levels of activated Akt (pAkt) (**Figure 2B,C**). The switch from ERK to Akt signaling resulted in cell growth that was resistant to the MAPK/ERK Kinase (MEK) inhibitor, PD98059, and more sensitive to the PI3K inhibitor, LY294002 (**Figure 2D**). These findings show that loss of cavin-3 shifts cellular signaling to an Akt-dominated state.

Activation of Akt can suppress apoptosis through inhibition of cytochrome c release from mitochondria and induction of inhibitor of apoptosis proteins (IAPs) (**Kennedy et al., 1999**; **Papapetropoulos et al., 2000**). To determine whether loss of cavin-3 suppresses apoptosis, we compared cell sensitivity to tumor necrosis factor-α (TNFα), an apoptosis-inducing cytokine. Treatment of SV589 fibroblasts with TNFα potently induced apoptosis as evidenced by the robust cleavage of the caspase-3 target, Poly [ADP-Ribose] Polymerase 1 (PARP1), and the prevalence of TUNEL positive cells (**Figure 2E**). Knockdown of cavin-3 progressively reduced cell sensitivity to TNFα in both assays. Resistance to TNFα correlated with induction of the IAP, survivin (**Figure 2B**), one of several antiapoptosis proteins induced by Akt (**Duronio, 2008**; **Guha and Altieri, 2009**). These findings show that loss of cavin-3 inhibits apoptosis.

Akt can also promote cell proliferation through activation of the mammalian target of rapamycin complex 1 (mTORC1) (**Manning and Cantley, 2007**). mTORC1 activates many biosynthetic pathways through phosphorylation and activation of S6K (**Duvel et al., 2010**). Loss of cavin-3 potently increased levels of phosphorylated S6K (pS6K) (**Figure 2B**), indicating that cavin-3 loss activates mTORC1. Microarrays comparing transcript profiles cells depleted of cavin-3 for 0, 3, 8, and 15 days

*Figure 3. Continued*

means ± SEM, n = 6. *p<0.05 as compared to No KD. H1299 cells serve as a positive control.

The following figure supplements are available for figure 3:

**Figure supplement 1**. Heat maps of gene transcripts involved in protein and nucleic acid biosynthesis.

showed that the rapid proliferation of 15-day knockdown cells was associated with substantial changes in gene transcription (*Figures 1A and 3A*). Many of transcripts that were up-regulated in the 15-day knockdowns are part of protein and nucleic acid biosynthetic pathways (*Figure 3B*). Cells fuel these biosynthetic pathways with metabolites derived from glycolysis. To increase metabolite levels, some rapidly dividing cells switch to aerobic glycolysis (Warburg metabolism), a metabolic state that is characterized by increased use of ferment-ative glycolysis even under normoxic conditions (*DeBerardinis et al., 2008*). Entry into aerobic glycolysis normally requires hypoxia induced factor 1 (HIF1), a transcription factor whose activity is controlled by the protein level of its α-subunit (HIF1α) (*Lunt and Vander Heiden, 2011*). Akt and mTORC1 increase HIF1α protein levels (*Schleicher et al., 2009*; *Duvel et al., 2010*) and knockdown of cavin-3 progres-sively increased both HIF1α levels and the use of fermentative glycolysis (*Figures 2B and 3C*). Glucose consumption and lactate production reached rates that were equivalent to those observed in H1299 cells, a non-small cell lung carcinoma cell line that lacks cavin-3 expression (*Xu et al., 2001*). These findings show that prolonged loss of cavin-3 activates the Akt/mTORC1/HIF1 pathway and induces aerobic glycolysis. Activation of these pathways likely facilitates rapid cell proliferation.

To test whether chronic loss of cavin-3 could recapitulate phenotypes observed in acute knockdown experiments, a germline knockout of the cavin-3 gene (*Prkcdbp*) was generated in mice (*Figure 4A*). As observed with 15-day knockdowns in SV589 cells, embryonic fibroblasts (MEFs) from cavin-3 knockout (Cavin-3 KO, *Prkcdbp*$^{-/-}$) animals displayed reduced levels of pERK, EGR1 and PTEN, elevated levels of survivin, pAkt, pS6K and HIF1α, increased rates of fermentative glycolysis, faster proliferation, heightened sensitivity to LY294002, resistance to PD98059 and insensitivity to TNFα as compared to wild-type MEFs (*Figure 4B–G*). Thus, genetic ablation of cavin-3 expression recapitulates phenotypes induced following long-term knockdown of cavin-3 by siRNA.

## Cavin-3 facilitates ERK activation by anchoring caveolae to F-actin of the membrane skeleton

To characterize the molecular mechanisms by which loss of cavin-3 alters cellular signaling, we first explored how cavin-3 facilitates ERK signaling. The loss of pERK but not pAkt that is associated with loss of cavin-3 (*Figures 2B and 4B*) suggested that cavin-3 acts downstream of mitogen receptors. Consistent with this conclusion, knockdown of cavin-3 did not inhibit EGF-dependent autophosphor-ylation of EGF receptors (*Figure 5A*). Furthermore, inhibition of protein phosphatase 2A (PP2A), which normally prevents the basal activity of mitogen receptors from activating downstream signaling cascades (*Wang et al., 2003*; *Van Kanegan et al., 2005*), was unable to activate ERK despite potent activation of Akt in cells depleted for cavin-3 (*Figure 5B*). These findings indicate that loss of cavin-3 disrupts signal transduction coupling between mitogen receptors and ERK.

Cavin-3 is localized in caveolae (*Bastiani et al., 2009*; *McMahon et al., 2009*), suggesting that the influence of cavin-3 on signaling involves caveolae. Caveolae have been implicated in both ERK and Akt signaling, though whether caveolae promote or suppress either pathway is not clear (*Liu et al., 1996*; *Mineo et al., 1996*; *Engelman et al., 1998*; *Furuchi and Anderson, 1998*; *Teixeira et al., 1999*; *Park et al., 2000*; *Cohen et al., 2003*). To determine whether cavin-3-dependent facilitation of ERK signaling involves caveolae, we used light and electron microscopy to test whether loss of cavin-3 influences caveolae abundance. We found that knockdown of cavin-3 both reduced caveolae abundance and re-localized the principal caveolae coat protein, caveolin-1, to the Golgi region (*Figure 5C*). The reduction in caveolae abundance correlated with loss of ERK responsiveness to EGF (*Figure 5D*). These findings indicate that cavin-3 is necessary for surface caveolae and suggest that caveolae facilitate signal transduction to ERK.

Caveolae associate with actin filaments at the plasma membrane (*Rohlich and Allison, 1976*; *Singer, 1979*) and disruption of F-actin impaired mitogen-dependent stimulation of ERK to the same extent as cavin-3 depletion (*Figure 5D*; *Aplin and Juliano, 1999*), suggesting that cavin-3 may increase caveolae abundance by anchoring caveolae to F-actin at the cell surface. Cavin-3 binds to the caveolar coat (*Bastiani et al., 2009*; *McMahon et al., 2009*); however, cavin-3 lacks a canonical actin-binding

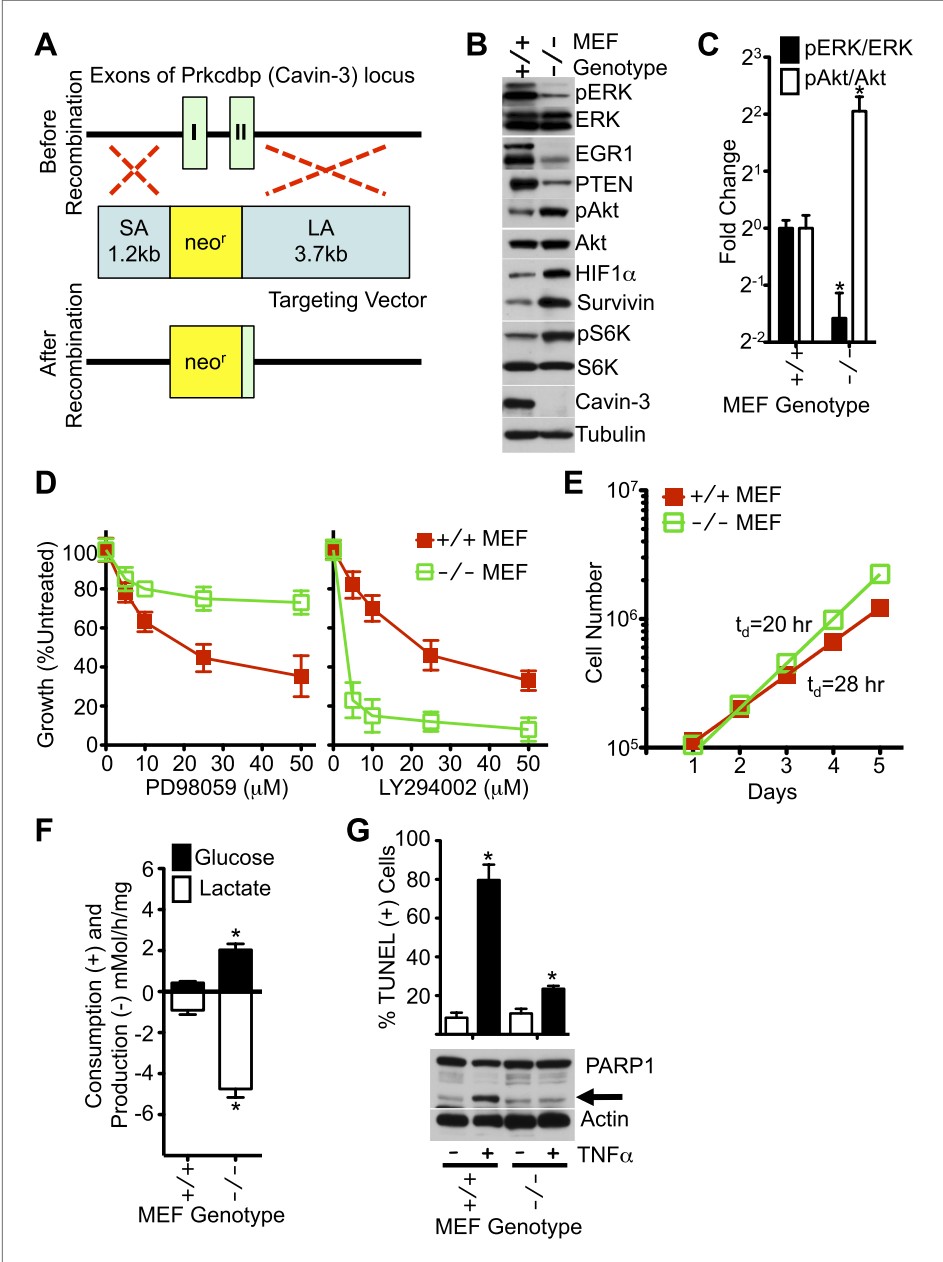

**Figure 4**. MEFs from Cavin-3 KO animals recapitulate phenotypes observed following long-term knockdown in human fibroblasts. (**A**) Diagram of the targeting strategy used to generate germline knockout of the Prkcdbp (cavin-3) gene. SA and LA indicate short arm and long arm regions of homology used for homologous recombination. Recombination replaced exon 1, most of exon 2 and the intron between the two coding exons with the neomycin resistance cassette. (**B**) Protein profiles of MEFs show that Cavin-3 KO MEFs have changes in protein distribution with respect to WT MEFs that are similar to the changes observed in human fibroblasts following 15-day knockdown. (**C**) Quantification of pERK and pAkt changes show that Cavin-3 KO MEFs have fourfold more pAkt and 3.7-fold less pERK than WT MEFs. Data are means ± SEM, n = 3. *p<0.05 as compared to WT MEFs. (**D**) Cavin-3 KO MEFs are more resistant to PD98059 and more sensitive to LY294002 than WT MEFs. Data are means ± SEM, n = 6. (**E**) Cavin-3 KO MEFs proliferate faster that WT MEFs. (**F**) Cavin-3 KO MEFs are more glycolytic than WT MEFs. Data are means ± SEM, n = 6. *p<0.05 as compared to WT MEFs. (**G**) Cavin-3 KO MEFs are more resistant to TNFα than WT MEFs. Arrow indicates cleaved PARP1. TUNEL data are means ± SEM from three independent experiments. *p<0.05 as compared to no TNFα. All assays were performed as in *Figures 1–3*.

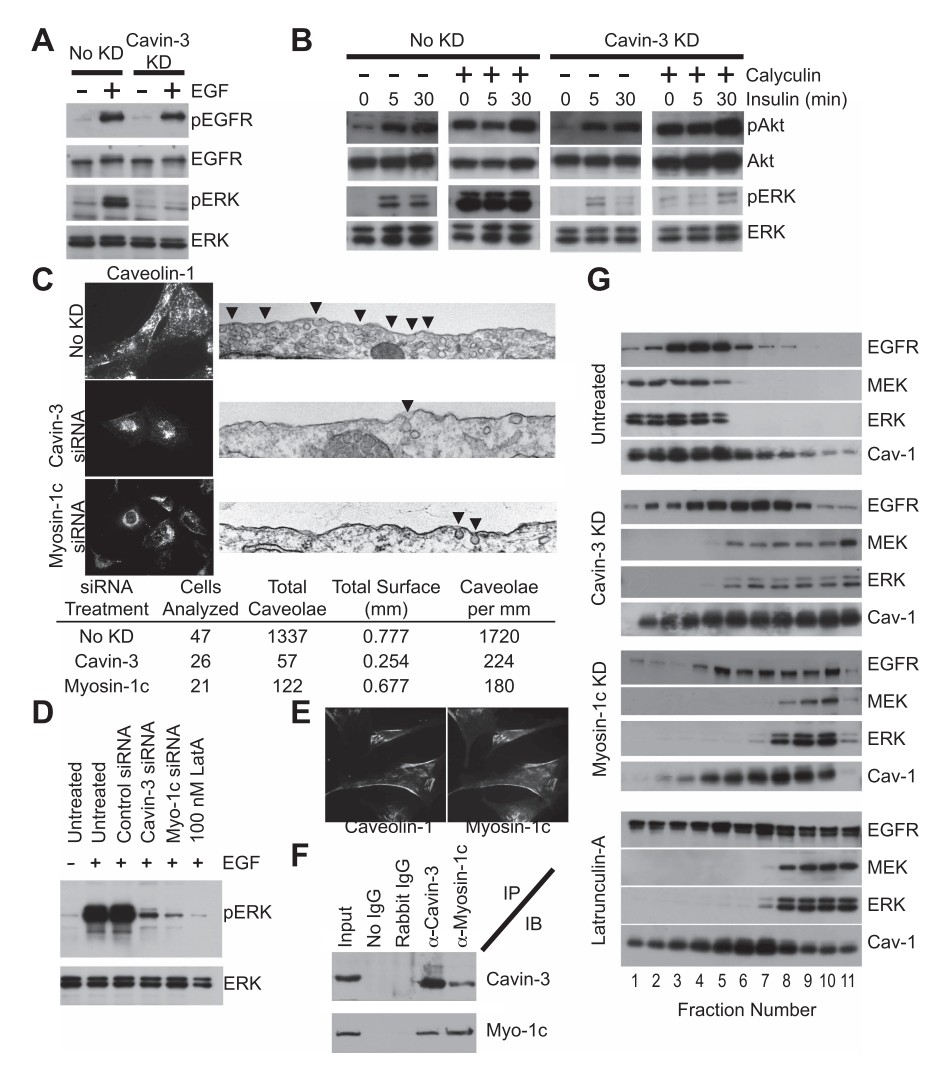

**Figure 5**. Cavin-3 anchors caveolae to F-actin via myosin-1c, thereby positioning MEK and ERK for activation by mitogen receptors. (**A**) 3-day knockdown of cavin-3 cripples signal transduction to ERK, but has little effect on receptor autophosphorylation. Fibroblasts (SV589) were mock treated or treated with cavin-3 siRNA, cultured in normal medium for 2 days, serum starved overnight, stimulated or not with 100 ng/ml EGF for 15 min, lysed and immunoblotted for the indicated protein. (**B**) Knockdown of cavin-3 prevents calyculin-A dependent activation of ERK. SV589 cells were treated or not with cavin-3 siRNA for 2 days, serum starved overnight, and treated or not with calyculin-A for 30 min followed by the addition of 100 nM insulin for the indicated times. Cell lysates were immunoblotted for the indicated protein. (**C**) 3-day knockdown of cavin-3 or myosin-1c redistributes caveolin-1 to the cell interior by immunofluorescence and reduces the abundance of morphological caveolae by thin section EM. Cells were treated or not with the indicated siRNA for 3 days and processed either for caveolin-1 immunofluorescence or thin section EM. Arrowheads indicate morphological caveolae. Quantification of caveolae abundance from 10 random fields was calculated as caveolae number per mm of plasma membrane length. Knockdown of cavin-3 or myosin-1 reduced caveolae abundance to similar extents (87% for cavin-3 siRNA and 90% for myosin-1c siRNA). (**D**) EGF-dependent activation of ERK requires cavin-3, myosin-1c and F-actin. Untreated, 3-day cavin-3 knockdown, 3-day myosin-1c knockdown or 30 min latrunculin-A treated cells were induced or not with EGF, lysed and immunoblotted for pERK and ERK. (**E**) Myosin-1c co-localizes with caveolin-1. Myosin-1c and caveolin-1 were localized by immunofluorescence. (**F**) Myosin-1c associates with cavin-3. SV589 fibroblasts were lysed and immunoprecipitated (IP) with the indicated antibody. Immunoprecipitants were immunoblotted (IB) for cavin-3 and myosin-1. (**G**) Co-fractionation of MEK and ERK with EGFR requires cavin-3, myosin-1c and F-actin. Cell membranes from untreated, 3-day cavin-3 knockdown, 3-day myosin-1c knockdown and 30 min latrunculin-A treated cells were isolated, shattered by sonication, separated on 5–30% Iodixanol gradients, fractionated and immunoblotted for EGFR, MEK, ERK, and Caveolin-1 (Cav-1).

site. To determine whether cavin-3 associates with an actin-binding protein, we tested candidate actin-binding proteins that had previously been identified in a proteomic screen as lipid raft/caveolae proteins (*Foster et al., 2003*) for their ability to satisfy the following five criteria: (i) co-localization with caveolin-1, (ii) reciprocal co-immunoprecipitation with cavin-3, (iii) knockdown that mislocalizes caveolin-1, (iv) knockdown that reduces surface caveolae and (v) knockdown that impairs ERK activation. Myosin-1c satisfied all five criteria (*Figure 5C–F*), indicating that myosin-1c participates in a cavin-3 linkage that anchors caveolae to peripheral actin.

To determine how caveolae might participate in ERK signaling, we reasoned that the role of caveolae should be downstream of ras because ras participates in both ERK and Akt activation. MEK and ERK have been shown to co-fractionate with mitogen receptors and caveolin-1 in cellular membranes of low protein-to-lipid density (*Liu et al., 1997*). Consistent with these published observations, we found that EGFR, caveolin-1, MEK and ERK co-fractionated in low-density membrane fractions isolated from untreated SV589 fibroblasts; however, knockdown of cavin-3, knockdown of myosin-1c or brief treatment with the actin-depolymerization compound, latrunculin-A, caused MEK and ERK to accumulate in the high-density fractions, which contain the bulk of cellular membrane protein (*Figure 5G*). The distribution of EGFR and caveolin-1 in the gradients became broader and heavier, but the majority of both EGFR and caveolin-1 remained in lighter fractions as compared to MEK and ERK. The EGFR remained on the plasma membrane because EGF treatment induced autophosphorylation of EGFRs in the absence of cavin-3 (*Figure 5A*). These observations indicate that a cytoskeletal linkage involving cavin-3, myosin-1c and F-actin positions MEK and ERK for activation by mitogen receptors, most likely by anchoring caveolae at the cell surface.

Cavin-3 associates with caveolae as part of a complex with caveolin-1 and other cavins (*Bastiani et al., 2009*; *McMahon et al., 2009*), suggesting that caveolin-1 and other cavins may participate in the cavin-3 linkage. Mammalian genomes encode four cavins of which two (cavin-1 and cavin-3) are readily detectable in fibroblasts grown under normal culture conditions. We tested whether cavin-1 and caveolin-1 are components of the cavin-3 linkage using immunoprecipitation with antibodies to myosin-1c and found that in addition to cavin-3 both cavin-1 and caveolin-1 co-precipitated with myosin-1c (*Figure 6A*). To test whether cavin-1 and caveolin-1 are necessary for the signaling function of the cavin-3 linkage, pERK and pAkt levels were compared in fibroblasts treated with siRNA against cavin-1, cavin-3, caveolin-1 or myosin-1c. Like the cavin-3 and myosin-1c knockdowns, knockdowns of caveolin-1 and cavin-1 increased pAkt at the expense of pERK (*Figure 6B,C*). However, unlike the myosin-1c knockdown, knockdown of cavin-1 reduced the protein level of cavin-3, while knockdown of caveolin-1 reduced protein levels of both cavin-1 and cavin-3. To test whether the effects of cavin-1 and caveolin-1 knockdowns on pERK and pAkt depend upon changes in cavin-3 protein levels, we over-expressed cavin-3 in SV589 fibroblasts (*Figure 6D*) and repeated the four knockdowns. Cavin-3 over-expression partially protected cavin-3 levels from knockdown of cavin-1, cavin-3 or caveolin-1. This protection muted the effects of these knockdowns on pERK and pAkt levels (*Figure 6E,F*). Importantly, cavin-3 levels correlated better with pERK/ERK and pAkt/Akt ratios than did levels of either cavin-1 or caveolin-1 (*Figure 6C,F,G*). The strength of this correlation suggests that cavin-3 is the limiting component of the cavin/caveolin complex for caveolar influence on ERK and Akt signaling. Both caveolin-1 and cavin-1 are required for normal abundance of caveolae (*Drab et al., 2001*; *Razani et al., 2001*; *Hill et al., 2008*; *Liu et al., 2008*). Caveolin-1 is an integral membrane protein that polymerizes to form filaments that coat the cytosolic surface of caveolae (*Fernandez et al., 2002*). Cavin-1 has been proposed to serve as an adaptor that links other cavins to the caveolin coat (*Bastiani et al., 2009*). We propose that cavin-3 promotes efficient signal transduction to ERK by bridging between the cavin-1/caveolin-1 complex and myosin-1c (*Figure 6H*).

## Stable expression of cavin-3 in cancer cells increases caveolae abundance coincident with suppression of Akt signaling, Warburg metabolism and resistance to apoptosis

Loss of cavin-3 is common in cancer cells (*Xu et al., 2001*; *Zochbauer-Muller et al., 2005*; *Lee et al., 2008*; *Martinez et al., 2009*; *Tong et al., 2010*; *Caren et al., 2011*; *Lee et al., 2011*) and many cancer cells show elevated Akt/mTORC1 signaling, aerobic glycolysis and resistance to apoptosis (*Manning and Cantley, 2007*; *DeBerardinis et al., 2008*; *Duronio, 2008*). To test whether loss of the cavin-3 linkage is necessary for the altered cell signaling, metabolism and apoptosis phenotypes of cancer cells we tested whether reconstitution of the cavin-3 linkage was sufficient to normalize ERK/Akt

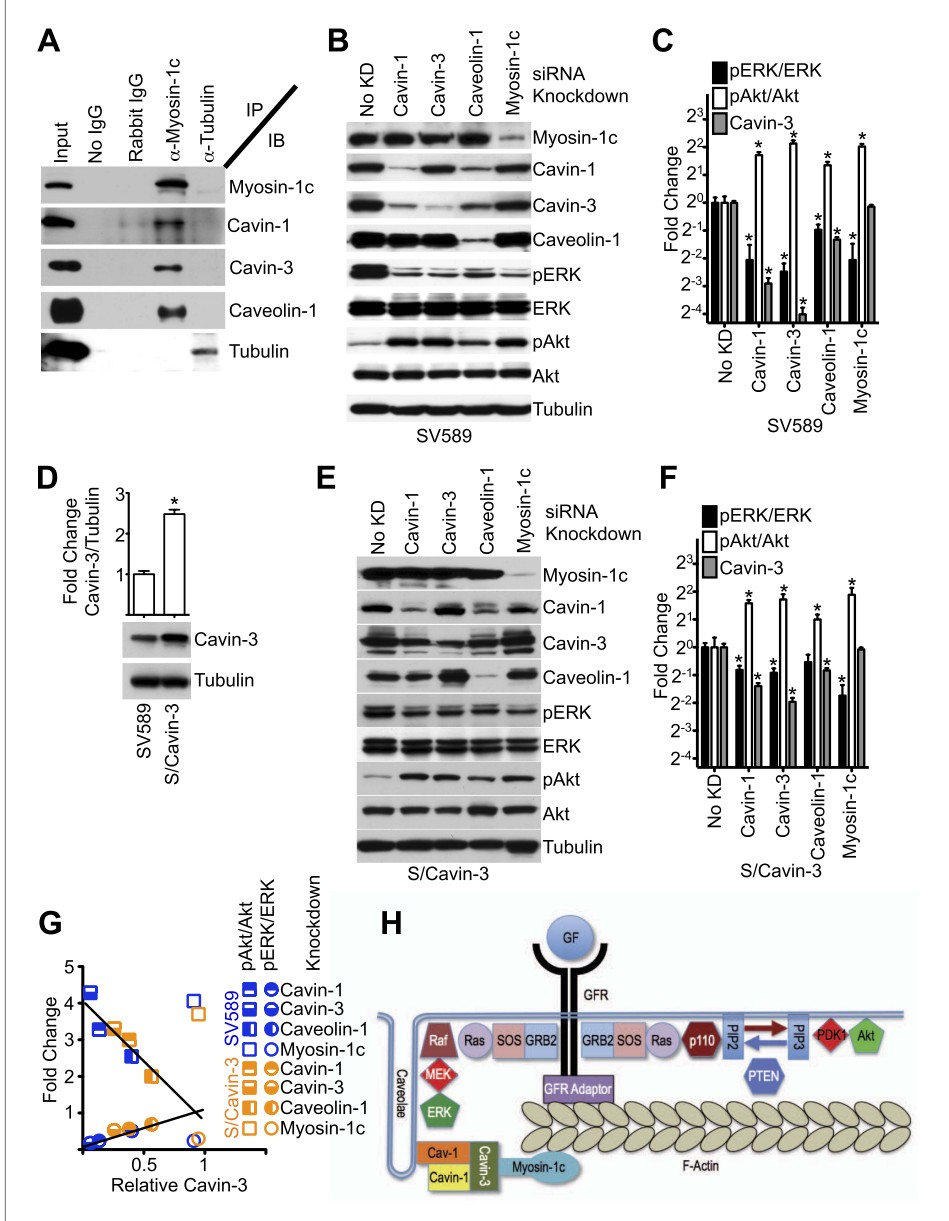

**Figure 6**. Linkage function requires myosin-1c, cavin-1, cavin-3, and caveolin-1. (**A**) The cavin-3 linkage involves caveolin-1, cavin-1, and myosin-1c. SV589 fibroblasts were lysed, immunoprecipitated (IP) with the indicated antibody and immunoblotted (IB) for the indicated protein. (**B**) Knockdown of cavin-1, cavin-3, caveolin-1 or myosin-1c suppresses pERK and augments pAkt levels. SV589 fibroblasts were treated with siRNA against cavin-1, cavin-3, myosin-1c or caveolin-1 for 3 days and immunoblotted for the indicated proteins. (**C**) Quantification of the effects of knockdowns on pERK/ERK and pAkt/Akt levels. Data are means ± SEM, n = 3. *p<0.05 as compared to No KD. (**D**) Stable over-expression of cavin-3 in SV589 fibroblasts (S/Cavin-3 cells) increases cavin-3 levels 2.5-fold over parental SV589 cells. (**E**) Over-expression of cavin-3 mutes effects of cavin-1, cavin-3, and caveolin-1 siRNAs on pERK and pAkt levels. S/Cavin-3 cells were treated with siRNA against cavin-1, cavin-3, caveolin-1 or myosin-1c for 3 days and blotted for the indicated proteins. The exposure of the cavin-3 blot was selected based upon similarity of the No KD controls in panels **B** and **E**. (**F**) Quantification of the effects of knockdowns on pERK/ERK and pAkt/Akt levels. Data are means ± SEM, n = 3. *p<0.05 as compared to No KD. (**G**) The mean values for pERK/ERK and pAkt/Akt for cavin-1, cavin-3, caveolin-1, and myosin-1c knockdowns from panels **C** and **F** were plotted against cavin-3 protein levels. Linear regression was performed on the six pERK/ERK data points and six pAkt/Akt data points for the cavin-1, cavin-3, and caveolin-1 knockdowns. R2 values for the pERK/ERK and pAkt/Akt lines are 0.937 and 0.930, respectively. (**H**) Model of the cavin-3 linkage between caveolae and F-actin in the context of the signaling pathways leading to ERK and Akt activation.

signaling, cell metabolism and apoptosis. To facilitate this reconstitution, we looked for a cancer cell line that lacks cavin-3, but expresses normal levels of all other linkage components. While many cancer cell lines lack cavin-3 (*Xu et al., 2001*; *Bastiani et al., 2009*; *Lee et al., 2011*), some lines such as PC-3 lacked additional linkage components (*Figure 7A*; *Bastiani et al., 2009*). H1299 cells are a line of non-small cell lung carcinoma cells that lacked detectable cavin-3, but exhibited levels of cavin-1, caveolin-1, myosin-1c and actin that were similar to the endogenous levels of SV589 fibroblasts (*Figure 7A*). Comparison of the pERK and pAkt responses to mitogens showed that H1299 cells had pERK and pAkt responses that were similar to SV589 fibroblasts following cavin-3 knockdown (*Figure 7—figure supplement 1*). To reconstitute cavin-3 in H1299 cells, retroviral vectors were used to stably express either GFP alone (H/GFP) or GFP and cavin-3 (H/Cavin-3). Use of a weak promoter-expression system generated levels of cavin-3 expression in H/Cavin-3 cells that were similar to endogenous levels expressed in SV589 fibroblasts (*Figure 7B*). Re-expression of cavin-3 restored the cavin-3 linkage as evidenced by the ability of myosin-1c antibody to co-precipitate cavin-1, cavin-3, and myosin-1c from lysates of H/Cavin-3 cells, but not H/GFP cells (*Figure 7C*). Restoration of the cavin-3 linkage resulted in a 7.6-fold increase in surface caveolae and redistribution of caveolin-1 to the plasma membrane (*Figure 7D*). As compared to parental H1299 or H/GFP cells, H/Cavin-3 cells displayed higher levels of pERK and EGR1 (*Figure 7B,E*); lower levels of pAkt, HIF1α and pS6K (*Figure 7B,E*); and increased sensitivity to PD98059 and resistance to LY294002 (*Figure 7F*). These signaling changes correlated with reductions in the rate of cell growth, glucose uptake and lactate production (*Figure 7G,H*). Cavin-3 re-expression also suppressed survivin levels and sensitized cells to TNFα-dependent apoptosis (*Figure 7B,I*). These findings show that restoration of the cavin-3 linkage in cancer cells can normalize ERK/Akt signaling, cell metabolism and apoptosis sensitivity.

## Cavin-3 suppresses the Akt/mTORC1/HIF1 pathway through EGR1

Loss of EGR1 and PTEN paralleled the increase in pAkt (*Figures 2B and 4B*), suggesting that loss of cavin-3 drives Akt activation through loss of the pERK-EGR1-PTEN axis. To test this hypothesis, we examined whether ectopic expression of EGR1 could suppress pAkt in the absence of cavin-3. Retroviruses were used to stably express (i) GFP alone, (ii) GFP and EGR1 or (iii) GFP and cavin-3 in *Prkcdbp*⁻/⁻ (Cavin-3 KO) MEFs and H1299 cells. Expression of EGR1 in either Cavin-3 KO MEFs or H1299 cells suppressed pAkt levels to the same extent as expression of cavin-3 (*Figure 8A,B*). These observations indicate that EGR1 acts downstream of cavin-3 to suppress Akt activation. Interestingly, while expression of either EGR1 or cavin-3 restored PTEN expression to a normal level in Cavin-3 KO MEFs, expression of neither EGR1 nor cavin-3 substantially improved PTEN expression in H1299 cells despite potent suppression of pAkt. The PTEN promoter in H1299 cells is hypermethylated (*Soria et al., 2002*) and this methylation may limit the ability of EGR1 to drive PTEN expression. The ability of cavin-3 and EGR1 to nonetheless suppress Akt activation indicates that EGR1 suppresses Akt activation through mechanisms that are independent of PTEN protein level.

Expression of EGR1 was sufficient to suppress aerobic glycolysis in both Cavin-3 KO MEFs and H1299 cells (*Figure 8*). EGR1 expression suppressed both pS6K and HIF1α levels in both cell lines (*Figure 8A*) and loss of HIF1α correlated with reductions in glucose consumption and lactate production (*Figure 8C*). Akt and mTORC1 induce HIF1α and the ability of EGR1 to suppress Akt activation indicates that loss of cavin-3 induces aerobic glycolysis via loss of EGR1-dependent suppression of the Akt/mTORC1/HIF1 pathway.

In contrast to the effects of EGR1 on cell metabolism, only cavin-3 re-expression was able to rescue sensitivity to TNFα (*Figure 8D*), indicating that cavin-3 supports an EGR1-independent process that is necessary for TNFα-sensitivity. Expression of EGR1 also did not restore pERK levels (*Figure 8A,B*) or drive caveolin-1 to the plasma membrane (*Figure 8E*). Active ERK facilitates apoptosis through both the intrinsic and extrinsic pathways (*Cagnol and Chambard, 2010*) and the ability of cavin-3 to support normal apoptosis sensitivity may require both the EGR1-dependent reduction in pAkt and a caveolae-dependent increase in pERK. Together, these findings show that cavin-3 activates at least two processes: (i) an EGR1-dependent process that suppresses the Akt/mTORC1/HIF1 pathway; and (ii) an EGR1-independent process that is necessary for normal apoptosis.

## Loss of cavin-3 in vivo causes cachexia

The signaling changes that were observed in cell culture following loss of cavin-3 were also observed in vivo. Lung tissue from *Prkcdbp*⁻/⁻ (Cavin-3 KO) animals showed decreased pERK, EGR1, and PTEN

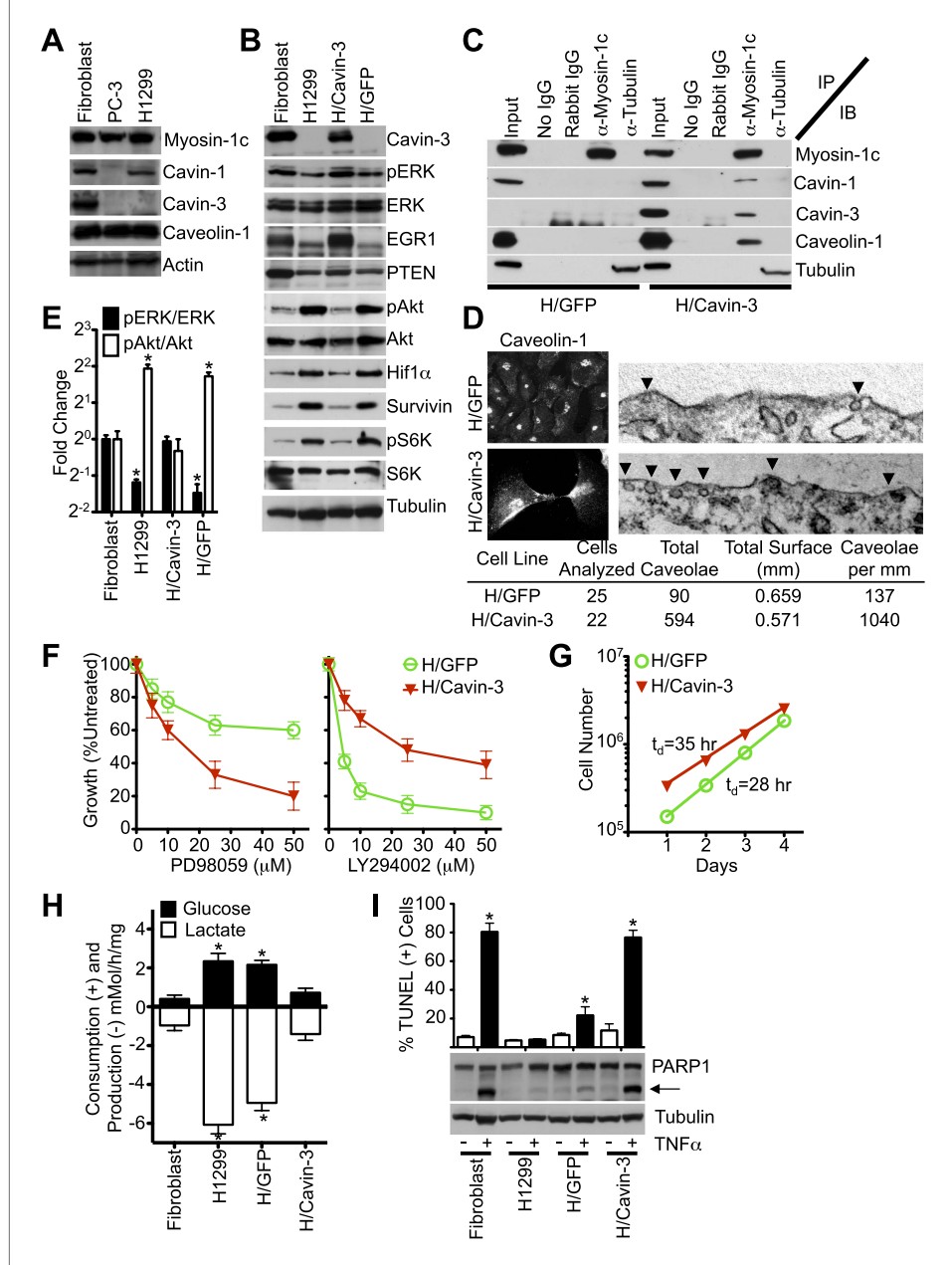

**Figure 7**. Stable expression of cavin-3 normalizes multiple phenotypes in cancer cells. (**A**) Comparison of cavin-1, cavin-3, myosin-1c, and caveolin-1 in SV589 fibroblasts (Fibroblast), PC-3 cells and H1299 cells. Representative immunoblots of the indicated proteins are shown. Comparison of pERK and pAkt responses to mitogen stimulation is provided in Supplement 1. (**B**) Expression of cavin-3 in H1299 cells reverts cellular signaling. Cell lysates from human SV589 fibroblasts (Fibroblast), H1299 cells, H1299 stably expressing cavin-3 (H/Cavin-3 cells) and H1299 stably expressing GFP (H/GFP cells) were immunoblotted for the indicated proteins. (**C**) Association of myosin-1c with cavins and caveolin-1 requires cavin-3. The indicated antibodies (IP) were used for immunoprecipitation from the indicated cells and immunoblotted for the indicated proteins (IB). (**D**) Cavin-3 expression increases caveolin-1 staining at the plasma membrane and increases the abundance of surface caveolae by 7.6-fold. (**E**) Quantification of changes in pERK/ERK and pAkt/Akt shows that expression of cavin-3 normalizes pERK and pAkt levels. Densitometry was performed on three replicates of the immunoblots from panel **B**. Data are means ± SEM. *p<0.05 as compared to SV589 fibroblasts. (**F**) Cavin-3 expression increases sensitivity to PD98059 and decreases sensitivity to LY294002. Data are means ± SEM, n = 6. (**G**) Cavin-3 expression decreases proliferation rate. (**H**) Cavin-3 expression suppresses glycolysis. Data are means ± SEM, n = 6. *p<0.05 as compared to SV589 fibroblasts.
*Figure 7. Continued on next page*

*Figure 7. Continued*

(I) Cavin-3 expression sensitizes cells to TNFα. Arrow indicates cleaved PARP1. TUNEL data are means ± SEM from three independent experiments. *p<0.05 as compared to no TNFα. All assays were performed as in *Figures 1–3*.
The following figure supplements are available for figure 7:

**Figure supplement 1**. pERK and pAkt responses to EGF or Insulin.

levels and increased pAkt and HIF1α levels as compared to lung tissue from wild-type animals (*Figure 9A,B*). The elevated HIF1α of Cavin-3 KO lung tissue was associated with increased fermentative glycolysis ex vivo (*Figure 9C*). Thus, in vivo loss of cavin-3 promoted Akt signaling at the expense of ERK and increased glycolytic metabolism. However, these signaling changes were not associated with developmental defects, as would be expected if apoptosis were compromised, or hyperplasia, as would be expected if cell proliferation were augmented. Cavin-3 KO mice did have shorter lifespan than control animals (*Figure 9D*) and the principal cause of death was cachexia as exemplified by a 40% reduction in body weight and severe lipodystrophy (*Figure 9E,F*). Lipodystrophy is frequently associated with hepatic steatosis (*Huang-Doran et al., 2010*) and areas of steatosis were observed in livers of Cavin-3 KO animals (*Figure 9G*). Despite the strong association of lung cancer with loss of cavin-3 expression (*Zochbauer-Muller et al., 2005*), we did not observe lung cancers and saw no differences in lung structure or alveolar density (*Figure 9G,H*). Masson's trichrome stain of lung sections also did not show differences in collagen fiber content, as would be expected for fibrosis (data not shown). A survey of additional tissues by H&E staining did not show differences between normal and Cavin-3 KO mice at either 4 months or 2 years of age (*Figure 9G* and data not shown). Thus, genetic ablation of cavin-3 expression increases Akt signaling at the expense of ERK, increases the use of fermentative glycolysis in tissues and causes cachexia, but is not sufficient to cause substantial de novo tumorigenesis.

## Discussion

The major finding of this study is that the tumor suppressor protein, cavin-3, controls the balance between ERK and Akt signaling with consequences for cell proliferation, metabolism, and apoptosis (*Figures 1–4, 7*). Cavin-3 promotes ERK signaling by anchoring the ERK activation module of caveolae to F-actin at the plasma membrane (*Figures 5 and 6*) and suppresses Akt signaling by promoting EGR1 and PTEN expression (*Figures 7 and 8*). The in vitro consequences of loss of cavin-3 include induction of Warburg metabolism, faster cell proliferation and resistance to apoptosis (*Figures 1–4, 7*). The in vivo consequences of loss of cavin-3 include elevated use of glycolysis and cachexia (*Figure 9*).

The cavin-3 dependent coupling of caveolae to F-actin is analogous to linkages that generate other specialized domains of the plasma membrane. Peripheral F-actin is crosslinked by spectrins into a viscoelastic network (the membrane skeleton) that associates with the plasma membrane through diverse linkages. The spatial organization of specific linkages allows the membrane skeleton to generate specializations within the plasma membrane and loss of different organizing linkages results in loss of specific specialized domains (*Bennett and Healy, 2008*). Our findings show that cavin-3 is part of a linkage necessary for the caveolae specialization (*Figures 5–7*). Intriguingly, insulin receptors also associate with F-actin and are enriched at the border between caveolae and the rest of the plasma membrane (*Foti et al., 2007*). The combination of a receptor-actin linkage with the cavin-3 linkage suggests that the skeleton assembles mitogen receptors with the caveolar ERK activation module to facilitate signal transduction coupling (*Figure 6H*). Other adaptors may link additional downstream signaling modules to the skeleton and different assemblies of receptors and signaling modules may dictate temporal, spatial, and cell-specific responses to common environmental cues. Skeleton-dependent integration of signaling pathways may promote cell differentiation because the sophistication of the skeleton correlates with the degree of cellular differentiation (*Bennett and Healy, 2008*) and defects in the skeleton are associated with dedifferentiation, increased cell proliferation and resistance to anoikis (*Mishra et al., 2005*; *Kumar et al., 2011*).

Whereas most linkages to the membrane skeleton are static, the cavin-3 linkage involves the motor molecule, myosin-1c, and the use of this motor may contribute to caveolae function in at least two ways. First, myosin-1c participates in the transport of lipid raft components (*Bond et al., 2013*) and the

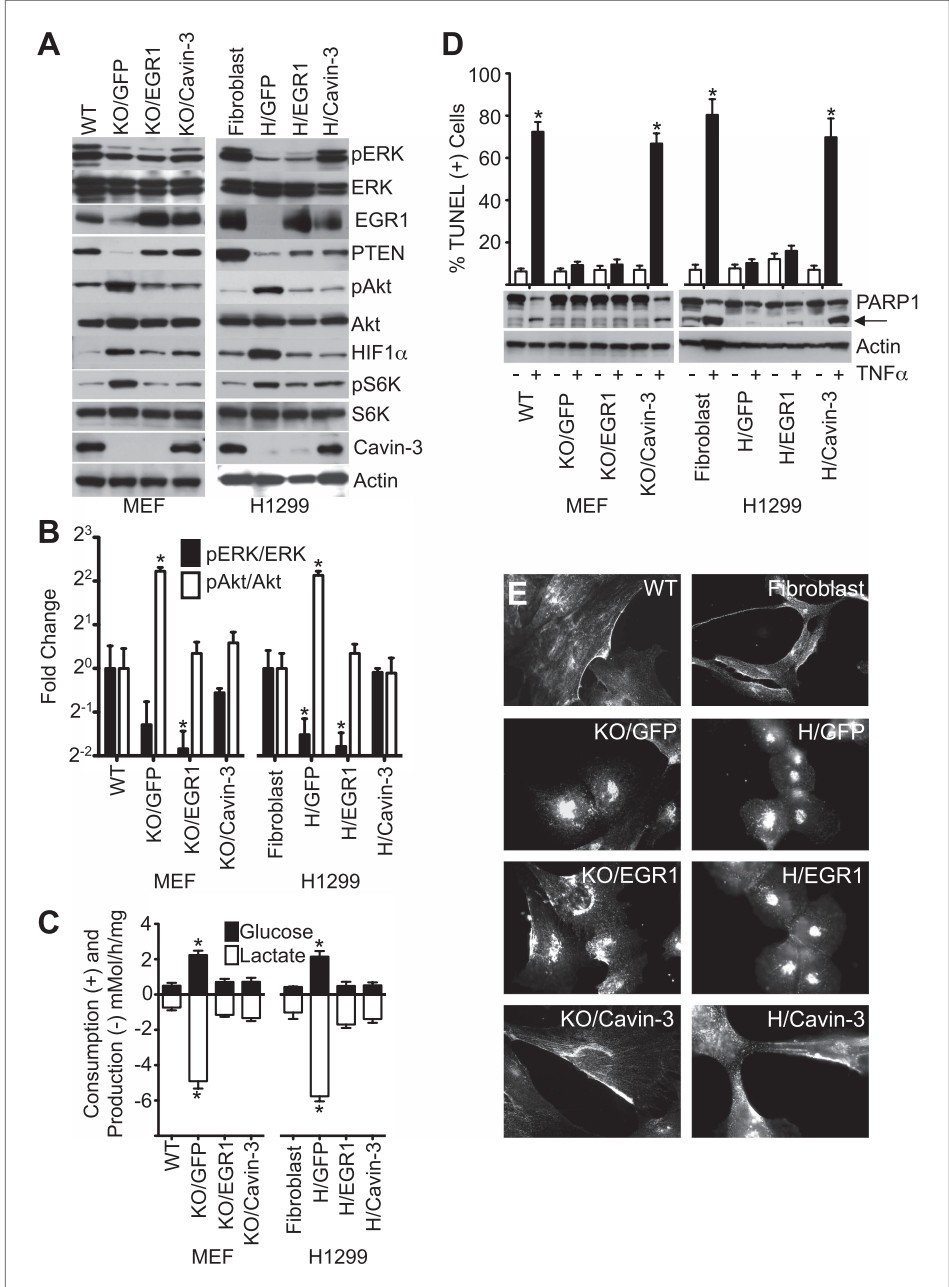

**Figure 8**. Loss of cavin-3 promotes Akt activation through loss of EGR1. (**A**) EGR1 expression is sufficient to suppress the Akt/mTORC1/HIF1α pathway. Immunoblotting of the indicated proteins was used to compare protein profiles of WT MEFs to Cavin-3 KO MEFs stably expressing GFP alone (KO/GFP), GFP and EGR1 (KO/EGR1) or GFP and cavin-3 (KO/Cavin-3) and human SV589 fibroblasts to H1299 cells stably expressing GFP alone (H/GFP), GFP and EGR1 (H/EGR1) or GFP and cavin-3 (H/Cavin-3). (**B**) Quantification of pERK/ERK and pAkt/Akt levels show that expression of EGR1 normalizes pAkt levels, but not pERK levels, in cavin-3 deficient cells. Data are means ± SEM, n = 3. *p<0.05 as compared to either WT MEFs (WT) or SV589 fibroblasts (Fibroblast). (**C**) Expression of EGR1 is sufficient to suppress aerobic glycolysis. Glucose uptake and lactate production data are means ± SEM, n = 6. *p<0.05 relative to WT MEF (WT) or SV589 fibroblast (Fibroblast) controls. (**D**) Expression of EGR1 is not sufficient to normalize TNFα-induced apoptosis. Arrow indicates cleaved PARP1. TUNEL data are means ± SEM, n = 3 experiments. *p<0.05 relative to cells not treated with TNFα. (**E**) Expression of EGR1 is not sufficient to normalize caveolin-1 distribution. Indicated cells were processed for caveolin-1 immunofluorescence. All assays were performed as in *Figures 1–3*.

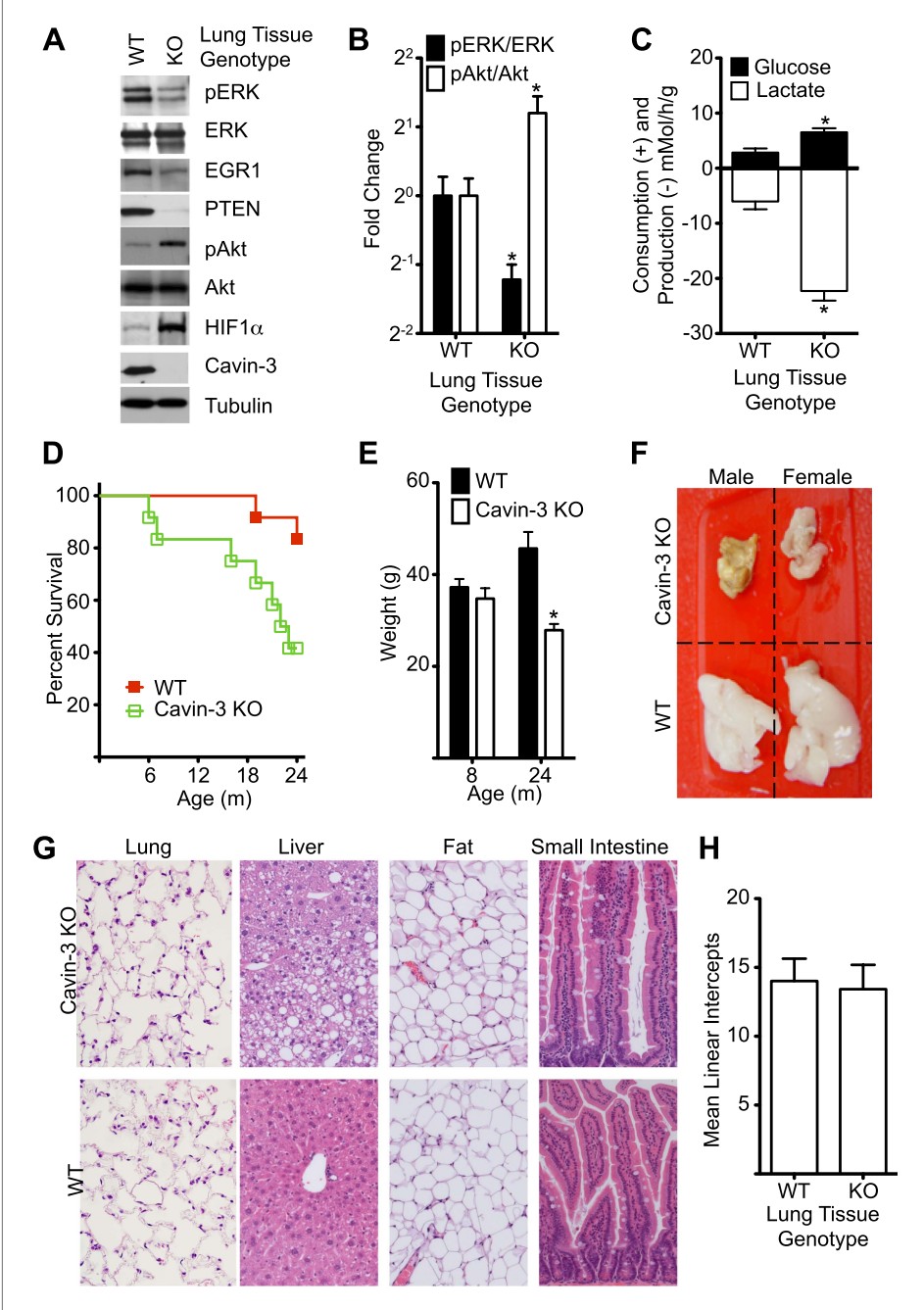

**Figure 9**. Cavin-3 KO animals have shortened lifespan resulting from late onset cachexia. (**A**) Protein profiles of lung tissue from 6-week old animals. (**B**) Quantification of pERK/ERK and pAkt/Akt in lung tissue from 6-week old animals show that loss of cavin-3 increases pAkt levels and decreases pERK levels by twofold. Data are means ± SEM, n = 6 (three males and three females). (**C**) Lung tissue from Cavin-3 KO animals is more glycolytic than normal. Lung tissue from 6-week old animals was excised and assayed for glucose consumption and lactate production over 4 hr in ex vivo culture. Data are means ± SEM, n = 6 (three males and three females). (**D**) Kaplan-Meier plot showing that Cavin-3 KO mice have decreased lifespan (n = 12). (**E**) Cavin-3 KO animals have a 40% reduction in body mass. *p<0.05 as compared to WT animals. (**F**) Cavin-3 KO animals have lipodystrophy. Shown are dissected abdominal fat pads. (**G**) H&E staining of abdominal fat pad, lung, liver, and small intestine. (**H**) Lung tissue from Cavin-3 KO animals does not show hyperplasia. Slides of WT and Cavin-3 KO lung sections were coded and imaged (four fields from six sections for WT and four fields from four sections for Cavin-3 KO) by a blinded observer. Mean linear intercepts of coded images were measured by a blinded observer. Data shown are means ± SD. p value is 0.6006.

use of myosin-1c may couple delivery of caveolar components with caveolae anchorage. Second, caveolae are endocytic structures and myosin-1c may control movement of caveolae vesicles within the cortical actin network. The majority of caveolae-derived vesicles remain at the cell periphery where they re-fuse with the plasma membrane in a cyclic process referred to as potocytosis (*Anderson et al., 1992*; *Boucrot et al., 2011*). Myosin-1c may drive caveolae-derived vesicles back to the cell surface in a calcium-regulated manner because calcium channels are enriched in caveolae (*Isshiki et al., 1998*; *Pani and Singh, 2009*), the Donnan effects generated by endocytosis promote calcium channel opening (*Saito et al., 2007*) and calcium promotes both myosin-1c motor activity (*Barylko et al., 1992*; *Zhu et al., 1998*) and recycling of caveolae-derived vesicles back to the cell surface (*Lin et al., 2012*). Potocytosis of caveolae may regulate the strength of signal transduction from surface receptors to ERK or allow ERK signal transduction to be moved to different sites along the plasma membrane.

Importantly, the effects of cavin-3 on ERK signaling are matched by inverse changes in Akt signaling. EGR1 can participate in a reciprocal relationship between ERK and Akt (*Yu et al., 2011*) and our findings show that cavin-3 suppresses pAkt via EGR1 (*Figure 8*). EGR1 induces PTEN expression (*Baron et al., 2006*); however, our findings show that induction of PTEN expression is not the only mechanism by which EGR1 suppresses Akt activation. How EGR1 suppresses Akt activation is currently under investigation. Alterations in cavin-3 protein levels have proportionate effects on pERK and pAkt levels (*Figure 6G*) and the tightness of this correlation may be the result of positive and negative feedback loops involving EGR1 and cavin-3. EGR1 can promote ERK activation (*Shen et al., 2011*) and ERK activation drives EGR1 expression (*Cohen, 1996*). This positive feedback loop requires cavin-3 because both pERK and EGR1 expression are suppressed in the absence of cavin-3 (*Figures 1–4, 7*). Cavin-3 and EGR1 also act within a negative feedback loop because EGR1 down-regulates cavin-1 and caveolin-1 expression (*Joshi et al., 2012*) and loss of either cavin-1 or caveolin-1 suppresses cavin-3 expression (*Figure 6*). Regulation within these feedback loops may set the balance point between pERK and pAkt.

Interestingly, modulation of cellular phenotypes in response to changes in this balance point involves substantial hysteresis. Treatment with cavin-3 siRNA reduces cavin-3 protein levels by more than 80% within 3-days; however, the phenotypic effects of cavin-3 depletion require 10–15 days to manifest fully (*Figures 1–3*). Many of these phenotypic effects involve extensive changes in gene expression (*Figures 1A and 3A*). While the extent of these cellular changes may require some time to complete, our data suggest that more active processes are responsible for the hysteresis. EGR1 protein levels fall slowly during the 15-day time course, despite rapid loss of pERK (*Figure 2B*) and the short half-life of EGR1 protein (*Waters et al., 1990*). EGR1 expression can be supported by means other than ERK (*Hallahan et al., 1991*; *Guillemot et al., 2001*) and these mechanisms may slow the loss of EGR1. The gradual loss of EGR1 coincides with gradual increases in survivin, pS6K and HIF1α (*Figure 2B*). The slowly evolving changes in these and other factors likely dictate the time-dependence for the manifestations of resistance to apoptosis, aerobic glycolysis and acceleration in cell proliferation. The length of time required to reach the final phenotypic state also provides a general note of caution with respect to knockdown experiments. siRNA knockdowns are typically assayed 3 days post-transfection; however, our data show that this time window may capture an intermediate state that differs substantially from chronic loss-of-function. Characterization of inter-mediate states may improve understanding of how proteins such as cavin-3 impact the panoply of cellular processes.

The ability of cavin-3 to influence cell signaling, proliferation, metabolism, and apoptosis provides explanations for how cavin-3 functions as a tumor suppressor; however, Cavin-3 KO animals do not show substantial increases in spontaneous cancers. These observations imply that loss of cavin-3 is not sufficient for tumorigenesis. Consistent with this conclusion, loss of cavin-3 expression is more prevalent in late-stage/high-grade cancers than in early-stage/low-grade cancers (*Lee et al., 2008*; *Caren et al., 2011*; *Wikman et al., 2012*). A potential clue as to the role of cavin-3 in cancer comes from the observation that 35–55% of glial, lung, gastric, ovarian, breast, and colorectal cancers show hypermethylation in their cavin-3 promoters (*Xu et al., 2001*; *Zochbauer-Muller et al., 2005*; *Lee et al., 2008*; *Martinez et al., 2009*; *Tong et al., 2010*; *Lee et al., 2011*). Methylation of the cavin-3 promoter also occurs normally in trophoblasts when they invade the endometrium during preg-nancy (*Grigoriu et al., 2011*). Trophoblast invasion shares many features with cancer cell metastasis (*Ferretti et al., 2007*) and the degree of methylation in the cavin-3 promoter correlates with both trophoblast invasion potential and cancer metastasis (*Wikman et al., 2012*; *van Dijk et al., 2012*). The

accelerated cell proliferation, induction of Warburg metabolism and resistance to apoptosis that results from loss of cavin-3 may facilitate the ability of invading cells to survive and proliferate in new environments and may thus provide strong selection for loss of cavin-3 function in cancer cells. Strong selection pressure for loss of cavin-3 in cancer cells is suggested by the observation that while 41% of primary non-small cell lung carcinomas show methylation of their cavin-3 promoters, 81% of these carcinomas (N = 93) lack detectable cavin-3 expression by immunohistochemistry (*Zochbauer-Muller et al., 2005*). Lung cancers are not commonly detected until late in disease progression and loss of cavin-3 may facilitate stage progression to metastatic disease.

The most apparent defect in Cavin-3 KO animals is cachexia as evidenced by a 40% reduction in weight and severe lipodystrophy (*Figure 9*). Lipodystrophies have also been noted in humans and animals lacking either cavin-1 or caveolin-1 (*Cao et al., 2008*; *Kim et al., 2008*; *Liu et al., 2008*; *Hayashi et al., 2009*; *Asterholm et al., 2012*) and the association of cavin-3 with cavin-1 and caveolin-1 (*Bastiani et al., 2009*) combined with the dependence of cavin-3 protein on cavin-1 and caveolin-1 (*Figure 6*) suggests that these lipodystrophies are caused by a common mechanism. Loss of cavin-3 linkage components may cause lipodystrophy through selective death in adipocytes (*Martin et al., 2012*); however, lipid mobilizing factors are elevated in the circulation of Cavin-3 KO animals (data not shown), suggesting that lipolysis is responsible for the loss of triglyceride stores. Cavin-3 KO animals exhibit increased use of fermentative glycolysis (*Figure 9*) and this increase may promote lipolysis for the purpose of clearing lactate. Glycolysis generates lactate and hepatocytes convert lactate back to glucose through the Cori cycle, which fuels the necessary gluconeogenesis via oxidative phosphorylation of fatty acids. Loss of caveolae increases flux through the Cori cycle as evidenced by the increased rates of lactate production, hepatic gluconeogenesis and adipocyte lipolysis in $Cav1^{-/-}$ (Caveolin-1 KO) mice (*Asterholm et al., 2012*). Whole body lactate production increases with age (*Wallace, 2005*) and loss of cavin-3 may exasperate lactate production to a point after which fatty acid demand for gluconeogenesis triggers lipolysis of triglycerides stored in adipose tissue. As lactate production continues to increase, host responses to lactate may drive cannibalization of protein from muscle, which in the absence of cancer is the source of most lactate production. Cancers that commonly elicit cachexia include lung, breast, and colorectal cancers (*Fox et al., 2009*) and these cancers frequently lack cavin-3 expression (*Xu et al., 2001*; *Zochbauer-Muller et al., 2005*; *Lee et al., 2011*). Thus, the absence of cavin-3 in tumors may predispose patients to the development of cancer-associated cachexia, a condition that is the immediate cause of death for more than 20% of all cancer patients (*Tisdale, 2002*).

Recent work has shown that caveolae of different tissues and cell types have different cavin compositions (*Bastiani et al., 2009*; *Hansen et al., 2013*). The fibroblasts and epithelial cells examined here express cavin-1 and cavin-3, but lack cavin-2 and cavin-4. By contrast, adipocytes abundantly express cavin-2, myocytes have abundant cavin-4 and endothelial cells have different compositions depending upon tissue localization (*Stan et al., 1999*; *Ogata et al., 2008*; *Bastiani et al., 2009*; *Hansen et al., 2013*). Use of different cavins may provide alternative linkages that can support surface caveolae or provide novel functions for caveolae in different cell types. Many functions have been ascribed to caveolae including mitogen signaling, mechanosensing, nitric oxide signaling, endocytosis, and transcytosis (*Boscher and Nabi, 2012*; *Kiss, 2012*; *Mineo and Shaul, 2012*; *Nassoy and Lamaze, 2012*; *Parton and del Pozo, 2013*). Different cavin compositions may support different functions in different cellular settings. We show here that cavin-3 plays a critical role in the signal transduction function of caveolae and that cells, which normally express cavin-3, rely upon cavin-3 for normal ERK and Akt signaling with consequences for cell metabolism, apoptosis, and cell proliferation.

## Materials and methods

### Antibodies
The following antibodies were used in this study. Anti-EGFR; Anti-phospho-EGFR (pY1068); Anti-phospho-Erk1/2 (pT202/pY204); Anti-PTEN; Anti-phospho-Akt1 (pS473); Anti-Akt1; Anti-MEK1/2; Anti-survivin; Anti-PARP1; and Anti-c-Fos, (all from Cell Signaling, Danvers, MA); Anti-caveolin-1 (BD Transduction Laboratories, San Jose, CA); Anti-(human cavin-3) (Bethyl Labs, Montgomery, TX); Anti-(mouse cavin-3) (Proteintech Group, Inc., Chicago, IL); Anti-ERK1/2 (Millipore, Billerica, MA); Anti-HIF1α (Bethyl Labs); Anti-Cavin-1 (AbCam, Cambridge, MA); Anti-Myo1c (Santa Cruz Biotechnology, Santa Cruz, CA); Anti-Tubulin; and Anti-Actin (Sigma-Aldrich, St. Louis, MO).

## Biochemical reagents

Epidermal growth factor (MP Biomedicals); platelet-derived growth factor BB (Millipore); 12-O-Tetradecanoylphorbol-13-Acetate (Cell Signaling); Lysophosphatidic acid (Sigma); Recombinant human insulin (Sigma); tumor necrosis factor alpha (Cell Signaling) Latrunculin-A (Sigma); Glucose Assay Kit (Sigma); L-Lactate Assay Kit (Megazyme International, Wicklow, Ireland). Clean-Blot IP Detection Kit (Pierce Biochemicals, Rockford, IL) were purchased. siRNA against cavin-3 was from Dharmacon RNAi technologies (Thermo Scientific, Pittsburgh, PA) and consists of an equal molar mixture of the following three oligos: 5'-UGGCCAAGGCGGAGCGCGU, 5'-GCGGGAAGCUCCACGUUCU and 5'-GCACCGGAUUGCAGAAGGU. Each of the three oligos was individually active against cavin-3. siRNA against cavin-1 (sc-76293), myosin-1c (sc-44604) and caveolin-1 (sc-29241) were obtained from Santa Cruz Biotechnologies.

## Construction of mouse embryonic cell lines

Mouse fetuses were harvested from 14-day pregnant Cavin-3 KO and wild-type mice. Fetuses were removed from dissected uteruses, usually between 3–5 fetuses per uterus, and heads and liver of fetuses were removed and blood clots removed by washing with 5 ml sterile phosphate buffered saline. Approximately three fresh embryos were placed in a single 10 cm sterile culture dish and minced with a sterile single edge razorblade into 0.5–1 mm slices. Minced tissue was digested with 0.05% Trypsin-EDTA by adding 5 ml of Trypsin-EDTA solution to each culture dish and incubating at 37°C under 5% $CO_2$ for 20 min with periodic agitation. After trypsin treatment, tissue suspensions were homogenized by passage through a 10 ml pipet. Tissue suspensions were plated in DMEM (low glucose) + 10% FBS and incubated for 5 hr at 37°C/5% $CO_2$. Medium was then replaced with fresh medium and cells were passaged 1:3 every 4 days for six passages, when cells entered crisis. During crisis, cells were re-fed every 3 days and split 1:2 once per week (or when plates reached confluence) for 3 months. The results of the Cavin-3 KO MEF line shown in *Figure 4* are typical of derived Cavin-3 KO MEF lines.

## Cells and cell culturing

MEFs and the human fibroblast cell line, SV589 (*Yamamoto et al., 1984*), were grown in Delbecco's Modified Eagle Medium (DMEM) supplemented with 10% fetal bovine serum (FBS) at 37°C with 5% $CO_2$. The lung cancer cell line, H1299, was obtained from the American Tissue Culture Collection (Manassas, VA). H1299 cells were grown in RPMI-1640 growth medium supplemented with 10% FBS at 37°C with 5% $CO_2$. Cell number was counted using a Bright Line Hemocytometer (Hausser Scientific, Horsham, PA).

## siRNA tranfections

For transfection of siRNAs into cells, an RNA-liposome suspension was prepared for each sample by mixing 2 ml of serum-reduced OPTI-MEM I media (Gibco, Grand Island, NY) with 480 nmol of siRNA and 36 μl of Lipofectamine RNAiMAX Reagent (Invitrogen, Grand island, NY) followed by incubation at room temperature for 20 min. The suspension was added to a 10 cm dish prior to layering 10 ml of cell suspension atop the RNA suspension. Cell suspensions were prepared by trypsination and suspension in DMEM supplemented with 10% FBS. Final siRNA concentration was 40 nM. 250,000 cells per 10 cm dish were added. After 5 hr incubation with RNA suspension at 37°C, culture medium was replaced with fresh growth medium. For prolonged knockdown, the described siRNA treatment was repeated on days 5, 9, and 12 such that assays used cells 3 days post siRNA treatment at all time points, unless otherwise indicated in the figure legend.

## Cell treatments and SDS sample preparation

For all mitogen stimulation experiments, test cells were grown to near confluence, then serum starved for 20 hr prior to addition of serum deficient medium containing mitogens indicated in figure legends. Mitogens were used at the following working concentrations: EGF at 100 ng/ml; PDGF at 20 ng/ml; Insulin at 100 nM; TPA at 200 nM; and LPA at 20 μM. Cells that received Latrunculin-A were pretreated with this actin sequestering agent for 20 min at 25 nM final concentration prior to EGF stimulation. After mitogen treatment, cells were washed twice with PBS and gently scraped into 5 ml PBS supplemented with phosphatase and protease inhibitor cocktails (RPI Corporation, Mount Prospect, IL). Cells were recovered by centrifugation at 1,000 × *g* for 3 min and then resuspended in 1 ml PBS supplemented with protease and phosphatase inhibitor cocktails at 4°C. Protein concentrations were determined by Bradford Assay, cell suspensions were diluted to 1 mg/ml concentration in SDS Sample buffer and denatured by heating to 100°C for 10 min.

## Immunoblotting

Equal protein loads of SDS denatured whole cell lysates were resolved by SDS-PAGE, and transferred onto PVDF membranes (Millipore). PDVF membranes were blocked for 30 min with 5% non-fat dry milk, washed with PBS, and incubated with appropriate concentrations of primary antibody overnight at 4°C. Blots were then washed, incubated with anti-rabbit or anti-mouse secondary IgGs conjugated with HRP (Biorad, Hercules, CA) for 1 hr at room temperature, washed and visualized on film using the Pierce ECL Chemiluminescence Substrate Kit (Thermo Scientific, Pittsburgh, PA). Quantification was performed by densotometry. All experiments were performed at least three times with a representative experiment shown in figures.

## Glucose/lactate consumption/production

Cells were subcultured onto 6 cm plates at an initial plating density of 40% confluency in 3 ml DMEM supplemented with 10% FBS and allowed to proliferate for 24 hr in incubators at 37°C and 5% $CO_2$. Growth medium was replaced with 1.5 ml of fresh DMEM supplemented with 10% FBS and incubated at 37°C in 5% $CO_2$ for a further 8 hr. 1 ml of fresh medium and 1 ml of conditioned media from each plate were centrifuged at 10,000 × $g$ for 5 min to remove trace insoluble materials and assayed for Glucose and Lactate content, using Glucose (Sigma) and Lactate (Megazyme) Assay Kits.

## Caveolae fractionation

Cellular membranes from 4 × 15 cm confluent dishes of untreated, siRNA treated or Latrunculin treated cells were separated by density using published protocols (*Smart et al., 1995*).

## Immunoprecipitations

Immunoprecipitations were performed as follows. Cell lysates were prepared from two 15 cm confluent dishes of cells by washing twice with 10 ml ice-cold PBS followed by scraping of cells into 5 ml of PBS with protease and phosphatase inhibitor cocktails. Scrapped cells were pooled, pelleted at 700 × $g$, and resuspended in 5 ml of TETN/OG (25 mM Tris-HCl, pH 7.5; 5 mM EDTA, 150 mM NaCl, 1% triton X-100, 60 mM octylglucoside) supplemented with 10 mM $CaCl_2$. Cell suspensions were incubated on ice for 1 hr, with votex mixing every 15 min to lyse cells. Nuclei were removed by centrifugation at 2000 × $g$ for 5 min. The post-nuclear supernatant was divided into aliquots. One aliquot was TCA precipitated and resuspended in 0.2 ml SDS sample buffer and is designated 'Input'. The remaining aliquots were incubated with no antibody or 10 µg non-specific rabbit IgG, rabbit anti-cavin-3, rabbit anti-tubulin or rabbit anti-myosin-1c. Suspensions were incubated at 37°C for 1 hr with gentle mixing. 25 µl of a 50% slurry of Protein A/G beads (Santa Cruz Biotechnologies) was then added and incubated with gentle mixing for 4 hr at 4°C. Beads were pelleted at 2000 × $g$ and washed twice with TETN500 (TET + 500 mM NaCl), twice with TETN250 (TET + 250 mM NaCl), and twice with Tris/EDTA (10 mM Tris, pH 7.5, 5 mM EDTA). Final pellets were dried at 55°C for 1 hr, resuspended in 0.2 ml SDS sample buffer and boiled. Eluted material was separated from beads by centrifugation at 10,000 × $g$ for 5 min. 1/20 of each sample was resolved on 4–15% Polyacrylamide SDS gels, transferred to PVDF membranes and incubated with indicated primary antibodies overnight at 4°C. Detection of primary antibodies used the Clean-Blot IP Detection Kit (Pierce Biochemicals), which suppresses detection of the denatured IgG heavy and light chains of precipitating antibodies.

## Immunofluorescence

Cells were grown on circular 12 mm optical borosilicate glass coverslips in 12 well cell culture plates. siRNA treatments were conducted on glass coverslips with appropriate number of cells to reach a confluence of 60% after 3 d culture at 37°C and 5% $CO_2$. After culturing, coverslips were washed with PBS two times, fixed with 3% paraformaldehype on ice for 15 min and permeabilized with 0.1% Triton X100 on ice for 10 min. Coverslips were then blocked with 1% normal goat serum (NGS) diluted in PBS for 30 min followed by incubation with primary antibody diluted into PBS supplemented with 1% NGS for 1 hr at room temperature. Coverslips were washed three times with PBS supplemented with 0.1% NGS, incubated with Alexaflur conjugated secondary antibodies (diluted in PBS + 1% NGS) for 1 hr at room temperature, washed three times with PBS + 0.1% NGS, washed twice with PBS, stained with 150 nM DAPI (4′,6-diamidino-2-phenylindole) in PBS for 5 min, washed three times with PBS and mounted using Fluoromount-G (Southern Biotech, Birmingham, AL). Cell images were taken on a Zeiss AxioImager M1 fluorescent microscope.

## Electron microscopy

Whole cells were gently scraped off culture dishes, fixed with 2% glutaraldehyde (EM Sciences, Fort Washington, PA) in PBS at room temperature for 1 hr, post-fixed with 1% Uranyl acetate in PBS for 1 hr, embedded in K4M epoxy resin, sectioned, and viewed with a Tecnai G2 Spirit 120 kV transmission electron microscope.

## Stable expression of cavin-3 or EGR1

The wild-type cavin-3 or EGR1 was stably expressed in SV589 fibroblasts, H1299 cells and Cavin-3 KO MEFs using a retroviral system as previously described (*Zhao and Michaely, 2008*). Briefly, cDNAs for human cavin-3 and human EGR1 were subcloned into the pMX-IRES-GFP bicistronic retroviral vector (*Liu et al., 1997*). Cavin-3, EGR1 or vector control retroviral vectors were co-transfected with the pAmpho packaging vector (Clontech, Mountain View, CA) into 293T cells to produce infectious, replication-defective retroviruses. Recombinant retroviruses were used to infect H1299 cells, which are a metastatic human non-small cell lung carcinoma cell line that does not express detectable cavin-3 (*Xu et al., 2001*), SV589 fibroblasts and Cavin-3 KO MEFs. The IRES element allows both the gene of interest and GFP to be translated from the same mRNA and thus cells that express GFP also express the gene of interest following successful genomic integration of the virus. GFP positive cells were purified using two rounds of fluorescence activated cell sorting (FACS) with a BD FACSAria cell sorter (Becton Dickinson). Two rounds of sorting generated populations that were at least 96% GFP positive.

## Construction of the cavin-3 targeting vector

Genomic DNA clones of mouse strain 129Sv/J were obtained by PCR into the pCR II vector (Invitrogen). From clones #F7 (3.2 kb DNA fragment) and #R7 (4.1 kb DNA fragment), an aligned 7.0 kb DNA fragment covering the full *Prkcdbp* (Cavin-3) gene including both exon1 and exon 2 was assembled and confirmed by sequencing with the mouse genome database (MGI). A pJB1 cassette vector expressing the neomycin resistant gene (Neoʳ) flanked by two LoxP sites (a gift from Joachim Herz, UTSW) was used to construct the Cavin-3 targeting vector using three steps. (i) The 1.2 kb Avr II-Xho I fragment (short arm, SA) from genomic clone #F7 was subcloned into the Xba I-Xho I sites of the pBS2-SK vector (Stratagene, La Jolla, CA), then subcloned using Not I into pJB1 to generate pJB1/SA. (ii) The 3.7 kb Xho I fragment (long arm, LA) from the full length Cavin-3 clone was subcloned into the Xho I site of pJB1 to generate pJB1/LA. (iii) The 7.0 kb Bam HI-Pac I fragment cut from clone pJB1/SA4, was ligated with the 8.0 kb Bam HI-Pac I fragment cut from clone pJB1/LA9 to complete the Cavin-3 targeting vector construction.

## Screening embryonic stem cells and generation of Cavin-3 KO mice

The targeting vector construct was electroporated into J1 ES cells derived from 129Sv/J mice by the Transgenic Core Facility under the direction of Robert Hammer on our campus. 600 ES cell clones resistant to both G418 and gancyclovir were expanded and analyzed by PCR resulting in the identification of two *Prkcdbp⁺/⁻* clones. These cells were injected into C57BL/6 blastocysts, which produced 14 chimeras as assessed by coat color. Germline transmission was determined by coat color and by PCR using the following primer sets: Exon II-mRNA-Fwd1 (F1), 5′- CAGATCAGCCAGAGGATGAAG-3′, Exon II-mRNA-Rev1 (R1), 5′- GGTAGGTTGAGGAGGTTCTGG-3′, and neo-S3 (neoF1), 5′- CAGAGG CCACTTGTGTAGCGCC-3′. The 272 bp (F1/R1, wt) and 532 bp (neoF1/R1, KO) amplification products were verified by sequencing. The null allele was then backcrossed through eight generations onto the C57BL/6 background.

## Microarray

Cells were freshly harvested and total RNAs were immediately extracted using an RNeasy Mini kit (Qiagen, Valencia, CA) following the manufacturer's instructions. RNA quality was checked using Bioanalyzer Chip (Agilent, Santa Clara, CA) and gene expression data were obtained using HumanHT-12 v4 Expression BeadChip (Illumina, San Diego, CA) through the UTSW Microarray Core Facility on campus.

## Mean linear intercept assay

Mean linear intercepts were calculated using H&E sections of normal and Cavin-3 KO lung at 600 × magnification by a blinded observer using ImageJ software. Four measurements per section were

made using sections obtained from three animals per group. p values were obtained by two-tailed Student's *t*-test using GraphPad Prism 5.

## Ex vivo lung culture

Sedated animals were sacrificed by exsanguination. Lungs were then removed, sliced, weighed, and cultured in D-MEM with antibiotics for 4 hr. Glucose and lactate production were assessed as in the in vitro cell culture experiments.

## Apoptosis assays

Duplicate plates of cells were treated with 10 µg/ml cycloheximide alone or in combination with 10 ng/ml TNFα for 15 hr in a humidified, 37°C $CO_2$ incubator. Cells were then either processed as whole cell lysates for PARP1 analysis or processed as whole cells for TUNEL. For TUNEL assays cells were washed with PBS, scraped into PBS, washed, fixed with 4% paraformaldehyde (10 min on ice), washed with PBS and permeabilized with ethanol (70% ethanol 15 hr −20°C). TUNEL was performed using a TUNEL kit (#A23210; Invitrogen) with Pacific Blue conjugated anti-BrdU (#B35129; Invitrogen). FACS analysis of TUNEL samples used a BD LSR II Flow cytometer at 405 nm. 10000 cells were counted for each sample and fluorescent profiles quantified using FlowJo software. All apoptosis data are derived from three independent experiments.

## Acknowledgements

We thank Joachim Herz, Robert Hammer and the Transgenic Core Facility for assistance with the mouse knockout; the Live Cell Imaging Core Facility and EM Core Facility for assistance with IF and EM imaging; Ralph DeBerardinis, John Minna, Mike White, Jerry Shay, Dorothy Mundy, and Phil Scherer for helpful discussions; and Jason Hall, Christine Kusminski, Natasha Buxton, Moriah Scarbrough and Jessica Sudderth for technical assistance. The work was supported by NIH grants HL085218 (PM), CA130821 (LM), and GM052016 (RA), and performed in laboratories constructed with support from NIH grant C06 RR30414.

## Additional information

### Funding

| Funder | Grant reference number | Author |
| --- | --- | --- |
| National Institutes of Health | HL085218 | Peter Michaely |
| National Institutes of Health | CA130821 | Lopa Mishra |
| National Institutes of Health | GM052016 | Richard GW Anderson |

The funders had no role in study design, data collection and interpretation, or the decision to submit the work for publication.

### Author contributions

VJH, JW, Conception and design, Acquisition of data, Analysis and interpretation of data; PL, SP, HD, MS, Acquisition of data, Analysis and interpretation of data; LM, Analysis and interpretation of data, Drafting or revising the article; RGWA, Conception and design, Contributed unpublished essential data or reagents; PM, Conception and design, Analysis and interpretation of data, Drafting or revising the article

### Ethics

Animal experimentation: This study was performed in strict accordance with the recommendations in the Guide for the Care and Use of Laboratory Animals of the National Institutes of Health. All animals were handled according to approved Institutional Animal Care and Use Committee (IACUC) protocols (2011-0096 and 2011-0098) of the University of Texas Southwestern Medical Center at Dallas (AAALAC assurance of compliance # A3472-01). No survival surgeries were performed and every effort was made to minimize suffering associated with end-stage cachexia.

## Additional files

### Major datasets

The following datasets were generated:

| Author(s) | Year | Dataset title | Dataset ID and/or URL | Database, license, and accessibility information |
|---|---|---|---|---|
| Michaely P, Hernandez VJ, Weng J, Ly P, Pompey S, Dong H, Mishra L, Schwarz M, Anderson RGW | 2013 | Data from: Cavin-3 dictates the balance between ERK and Akt signaling | http://dx.doi.org/10.5061/dryad.4950k | Available at Dryad Digital Repository under a CC0 Public Domain Dedication contains microarray data for SV589 fibroblasts treated or not with Cavin-3 siRNA for 3, 8 or 15 days and grown in serum, serum starved, serum starved then treated with 100 ng/ml EGF for 1 hr, or serum starved then treated with 100 ng/ml for 3 hr. Provided data are averaged transcript levels and ratios between averaged transcript levels. |
| Michaely P, Hernandez VJ, Weng J, Ly P, Pompey S, Dong H, Mishra L, Schwarz M, Anderson RGW | 2013 | Cavin-3 dictates the balance between ERK and Akt signaling | GSE50982; http://www.ncbi.nlm.nih.gov/geo/query/acc.cgi?acc=GSE50982 | In the public domain at GEO: http://www.ncbi.nlm.nih.gov/geo/. |

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
