## [Decision Letter]

Thank you for sending your work entitled “Cavin-3 dictates the balance between ERK and Akt signaling” for consideration at *eLife*. Your article has been favorably evaluated by a Senior editor and 3 reviewers, one of whom is a member of our Board of Reviewing Editors.

The Reviewing editor and the other reviewers discussed their comments before we reached this decision, and the Reviewing editor has assembled the following comments to help you prepare a revised submission.

The reviewers found that your study of cavin-3 was novel and interesting. However, a number of questions arose during the review that will need to be addressed in a revised manuscript.

1) No RNAi sequences are described and it appears that, in each case, only a single sequence was used; no control RNAi was employed; and no rescue studies were performed. Moreover, control studies using RNAi on WT cells are missing (e.g., Figure 6).

2) The phenotype described indicates defects in growth factor-stimulated signaling pathways (e.g., Figure 1). However, in two instances, no overt stimulation was employed (e.g., Figures 2 and 7). This should be corrected.

3) The authors show that TNF causes cell death when added to human and murine fibroblasts. Others have found that TNF does not cause cell death in such cells unless combined with inhibition of NF-kB, protein synthesis or RNA synthesis.

4) The authors state that the IEC profile is “driven by ERK”. What is the evidence for this?

5) The authors present a number of criteria for identifying the interacting protein as myosin-1c. The implication of the presentation is that proteins other than myosin-1c were tested and rejected. Please explain.

6) How general is the mechanism that cavin-3 dictates the balance between ERK and pAkt signaling? It is argued that control of PTEN is responsible for the effects of cavin-3 on Akt regulation. This cannot be the entire story, since PC3 cells are genetically deficient in PTEN – there must be an alternative mechanism. Presumably this is why analysis of PC3 was left out for Figure 5. Some discussion is in order.

7) Cavin-3 knock-out mice have recently been reported by Hansen et al. (Nat. Commun. 4:1831. doi: 10.1038/ncomms2808). These authors demonstrate no effect on caveoli formation in endothelial tissues and also show that, similar to the current study, the loss of caveolin-1 or cavin-1 results in reduced cavin-3 protein levels. The authors should discuss how this paper relates to their own study.

8) The authors propose a model whereby cavin-3 forms a complex with cavin-1 and caveolin-1 that links to F-actin and growth signals via an interaction between cavin-3 and myosin-1c (Figure 4). The data presented demonstrates that myosin-1c can immunoprecipitate with all three proteins so it is unclear what the evidence is for cavin-3 directly interacting with myosin-1c.

9) Figure 4: siRNA knockdown of caveolin-1 or cavin-1 leads to loss of cavin-3 protein, correlating with reduced ERK phosphorylation and increased Akt phosphorylation. Complementation studies should be performed to test the proposed model. Are the changes in ERK/Akt signaling after caveolin-1/cavin-1 knock-down due to reduced cavin-3 protein levels, or due to loss of the complex?

---

## [Author Response]

We wish to thank the reviewers for their helpful comments. In response to these comments, the following data have been added to the resubmission: (1) Edge analysis of the IE microarray data. This analysis illustrates that depletion of cavin-3 suppresses expression of ERK-induced transcripts. (2) Knockdowns of cavin-1, cavin-3, caveolin-1 and myosin-1c in SV589 fibroblasts over-expressing cavin-3. Comparisons of pERK/ERK and pAkt/Akt vs cavin-3 level in this data set together with that of the prior data set using SV589 fibroblasts expressing endogenous levels of cavin-3 show that pERK/ERK and pAkt/Akt levels correlate with the level of cavin-3, but not the levels of cavin-1 or caveolin-1. This finding indicates that the cavin-3 component of the cavin/caveolin complex dictates the pERK/pAkt ratio. (3) Immunoprecipitation data using H/GFP and H/Cavin-3 cells. These data show that the association of cavin-1 and caveolin-1 with myosin-1c requires the presence of cavin-3. (4) Time courses of pERK/pAkt responses to EGF for SV589 fibroblasts, SV589 fibroblasts treated with cavin-3 siRNA, PC-3 and H1299 cells. (5) pS6K/S6K data for the knockdown, MEF and H1299 models. These data show that loss of cavin-3 activates mTORC1 and that cavin-3 normally suppresses mTORC1 via EGR1. (6) Glucose uptake and lactate production for EGR1 and cavin-3 reconstitutions in the MEF and H1299 models. These data show that cavin-3 suppresses aerobic glycolysis via EGR1. (7) TNFα sensitivity for the EGR1 and cavin-3 reconstitutions in the MEF and H1299 models. These data show that cavin-3 drives an essential process for TNFα-dependent apoptosis that is EGR1-independent. In order to accommodate the new data, the figures have been restructured. Of particular note is that Figure 1 of the original submission has been broken into 3 figures (Figures 1, 2 and 3). The writing in the manuscript has been altered to incorporate both the new data and specific critiques made by the reviewers. Responses to individual critiques are detailed below.

*1) No RNAi sequences are described and it appears that, in each case, only a single sequence was used; no control RNAi was employed; and no rescue studies were performed. Moreover, control studies using RNAi on WT cells are missing (e.g., Figure 6)*.

Response 1: siRNA sequences are now provided in the Methods. Initial experiments showed similar effects on ERK signaling by three different cavin-3 RNAi sequences. All experiments shown in the manuscript used an equal molar mixture of these three cavin-3 oligos.

Response 2: To validate that the effects of cavin-3 siRNA treatment were due to loss of cavin-3 as opposed to off-target effects, we chose to test whether loss of cavin-3 had similar effects in different model systems in which loss of cavin-3 resulted from different means: the MEF model uses a genetic knockout of the cavin-3 locus and the H1299 model is an epigenetic hypomorph of cavin-3. All three in vitro models showed similar effects on ERK/Akt signaling, cell proliferation, cell metabolism and apoptosis.

Response 3: Rescue experiments were performed in Figures 5 and 6 (Figures 7 and 8 in the resubmission) with the MEF and H1299 models. The original submission provided immunoblots, ERK/Akt quantification and caveolin-1 immunofluorescence data in cells stably expressing either EGR1 or cavin-3 under control of a viral promoter. In the resubmission, we now provide cell metabolism and apoptosis data for the rescue experiments in both the MEF and H1299 models. This additional data now show that cavin-3 expression is sufficient to fully normalize both H1299 and MEF models, but that EGR1 expression is only able to normalize Akt/mTORC1/HIF1 signaling and cell metabolism. EGR1 is unable to normalize sensitivity to TNFα. This new finding suggests that cavin-3 supports an EGR1-independent process that is necessary for TNFα-induced apoptosis. The EGR1-independent process may involve caveolae or pERK, neither of which are normalized by EGR1 expression.

Response 4: Figure 6 (Figure 8 in the resubmission) does not contain an siRNA experiment. This figure details the viral expression of EGR1 and cavin-3 in Cavin-3^-/-^ MEFs and H1299 cells, neither of which express cavin-3 or EGR1 natively. The WT MEFs and SV589 fibroblasts are positive controls for the viral expressions. The writing in the Results has been changed to better indicate this point.

*2) The phenotype described indicates defects in growth factor-stimulated signaling pathways (e.g., Figure 1). However, in two instances, no overt stimulation was employed (e.g., Figures 2 and 7). This should be corrected*.

We began this study looking for an effect of cavin-3 loss on the EGF signaling pathway because of the importance of EGFR signaling in many carcinomas; however, results of Figure 1 show that loss of cavin-3 suppresses the ability of multiple mitogens individually and collectively (serum) to drive ERK activation. This finding spurred us to focus on cavin-3 function in the presence of serum. We have altered the wording throughout the manuscript to better reflect this focus.

*3) The authors show that TNF causes cell death when added to human and murine fibroblasts. Others have found that TNF does not cause cell death in such cells unless combined with inhibition of NF-kB, protein synthesis or RNA synthesis*.

The methods detailing the apoptosis assays were inadvertently left out. All apoptosis assays used cycloheximide. The Methods section now describes both the TUNEL and PARP1 cleavage assays in detail and the use of cyclohexamide is now indicated in the figure legends.

*4) The authors state that the IEC profile is “driven by ERK”. What is the evidence for this*?

The full statement was “Suppressed transcripts fell into two categories: those that were fully suppressed by three days of knockdown and those that fell gradually during the 15-day time course. Most IE response transcripts in the first group encoded proteins whose expression is driven by ERK.” This statement was based upon microarray data and the immunoblot in Figure 1. These observations showed that ERK-induced transcripts were selectively suppressed following cavin-3 knockdown. We now provide edge analysis in Figure 1 showing the 15 transcripts with the highest induction in the absence of knockdown together with their relative induction following 3, 8 and 15 days of knockdown. A literature search identified 8 of the 15 transcripts as genes whose transcription is promoted by ERK. All eight fail to be induced by EGF following 3 or more days of cavin-3 knockdown. References for ERK-dependent induction of these transcripts are provided.

*5) The authors present a number of criteria for identifying the interacting protein as myosin-1c. The implication of the presentation is that proteins other than Myosin1c were tested and rejected. Please explain*.

The caveolae/lipid raft associated actin binding proteins that were identified in the Foster paper were α-catenin, cofilin, ezrin, talin, myosin heavy chain, myosin-1a, myosin-1c, myosin-X and myosin-XV. Because this proteomic screen did not distinguish between caveolae and lipid rafts, we used literature data to exclude proteins that did not co-localize with caveolae. Myosin heavy chain forms filaments that pull on actin stress fibers, but caveolae are not associated with actin stress fibers. Myosin-X and myosin-XV are localized to filopodia, but caveolae are not. Myosin-1a is localized to brush borders, but caveolae are not. Talin is localized to cell adhesions, but caveolae are not. α-catenin participates in actin bundling but does not show strong caveolae co-localization. Cofilin is an actin disassembly factor. Tossing these proteins left myosin-1c and ezrin. Both proteins co-localized with caveolae, but only knockdown of myosin-1c was able to phenocopy the effects of cavin-3 knockdown on pERK. Subsequent experiments (co-IP, cell fractionation) validated a role for myosin-1c in the cavin-3 linkage.

*6.1) How general is the mechanism that cavin-3 dictates the balance between ERK and pAkt signaling*?

There are undoubtedly other mechanisms that influence the balance between ERK and Akt. First, loss of cavin-3 suppresses pERK levels ∼4-fold but does not eliminate pERK. Second, genetic loss of both ERK1 and ERK2 is lethal, but loss of cavin-3 is not. Third, not all cells that elicit pERK responses have caveolae. For example neurons and hepatocytes have few caveolae, but are able to mount ERK responses to trophic factors. Fourth, not all caveolae are equivalent. Caveolae of adipocytes have abundant cavin-2, myocyte caveolae have high levels of cavin-4 and endothelial caveolae are varied in their composition. What our data says is that for cells that normally express cavin-3, cavin-3 is necessary for a normal pAkt/pERK ratio with consequences for cell growth, metabolism and apoptosis. Of key significance is that epithelial cells appear to be a cell type that uses cavin-3 for this purpose and we show that reconstitution of cavin-3 in carcinoma cells that have lost cavin-3 expression is sufficient to normalize Akt/ERK signaling, cell metabolism, cell growth and sensitivity to TNFα. Given that loss of cavin-3 expression is widespread in cancer cells, loss of cavin-3 linkage function may be a common mechanism by which cancer cells achieve an Akt-dominated state that exhibits Warburg metabolism, rapid cell proliferation and resistance to apoptosis. We are beginning to look at other cell types to address the generality of the role we observe for cavin-3 in ERK and Akt signaling. The last paragraph of the Discussion now discusses the generality of our results.

*6.2) It is argued that control of PTEN is responsible for the effects of cavin-3 on Akt regulation. This cannot be the entire story, since PC3 cells are genetically deficient in PTEN – there must be an alternative mechanism. Presumably this is why analysis of PC3 was left out for Figure 5. Some discussion is in order*.

The reviewers are absolutely correct about additional mechanisms. H1299 cells suppress PTEN through promoter methylation and neither cavin-3 nor EGR1 was capable of substantially increasing PTEN levels, yet both cavin-3 and EGR1 normalized pAkt levels. We hypothesize that EGR1 drives expression of other proteins that inhibit Akt activation and are starting to explore how this suppression works using candidates identified in our microarray data. Regarding PC-3 cells, we chose not to pursue this cell line because these cells lack cavin-1 (Figure 7 in the resubmission) and loss of cavin-1 suppresses cavin-3 levels (Figure 6). PC-3 cells are of interest to us for future experiments both because these cells lack all four cavins, which will facilitate isogenic reconstitution experiments, and because caveolin-1 promotes cell migration and metastasis in PC-3 cells. Loss of cavin-3 is associated with both cancer cell metastasis and trophoblast migration and loss of cavin-3 may drive caveolin-1 into a process that facilitates cell migration. A discussion of additional mechanisms beyond PTEN protein level is now provided in both the Results and Discussion.

*7) Cavin-3 knock-out mice have recently been reported by Hansen et al. (Nat. Commun. 4:1831. doi: 10.1038/ncomms2808). These authors demonstrate no effect on caveoli formation in endothelial tissues and also show that, similar to the current study, the loss of caveolin-1 or cavin-1 results in reduced cavin-3 protein levels. The authors should discuss how this paper relates to their own study*.

The Hansen paper is now referenced and discussed.

*8) The authors propose a model whereby cavin-3 forms a complex with cavin-1 and caveolin-1 that links to F-actin and growth signals via an interaction between cavin-3 and myosin-1c (Figure 4). The data presented demonstrates that myosin-1c can immunoprecipitate with all three proteins so it is unclear what the evidence is for cavin-3 directly interacting with myosin-1c*.

We now provide immunnoprecipitation experiments in Figure 7 that show that cavin-3 is required for the ability of cavin-1 and caveolin-1 to co-precipitate with myosin-1c. This dependence coincides with cavin-3-dependent normalization of caveolae abundance, pERK, pAkt, pS6K, EGR1, PTEN, HIF1α, survivin, growth rate, glycolysis and sensitivity to PD98059, LY294002 and TNFα. While we recognize that direct tests of interaction using purified proteins would have been preferable, these experiments are not currently feasible. Cavin-3 is a phosphoprotein and we now have preliminary data showing that phosphorylation is required for cavin-3 function. We have identified two serines that must be phosphorylated for cavin-3 function and prediction algorithms suggest that an additional three residues may also be phosphorylated. Myosin-1c binds to negatively charged cargos and we suspect that these phosphorylations provide the complementary charges that drive myosin-1c interaction with cavin-3; however, until we know which residues must be phosphorylated and whether aspartate/glutamate substitutions can substitute for phosphorylation we cannot begin to test for direct interactions with purified proteins.

*9) Figure 4: siRNA knockdown of caveolin-1 or cavin-1 leads to loss of cavin-3 protein, correlating with reduced ERK phosphorylation and increased Akt phosphorylation. Complementation studies should be performed to test the proposed model. Are the changes in ERK/Akt signaling after caveolin-1/cavin-1 knock-down due to reduced cavin-3 protein levels, or due to loss of the complex*?

Complementation studies are now provided in Figure 6. We over-expressed cavin-3 by 2.5-fold in human fibroblasts and repeated the knockdown comparisons of pERK/ERK and pAkt/Akt. We find that over-expression of cavin-3 hindered the ability of cavin-1 and caveolin-1 knockdowns to suppress cavin-3 levels with commiserate effects on pERK/ERK and pAkt/Akt levels. Myosin-1c knockdown by contrast had little effect on cavin-1, cavin-3 or caveolin-1 level despite near maximal effect on pERK and pAkt levels. These observations indicate that cavin-3 and myosin-1c provide limiting activities that are necessary for normal ERK and Akt signaling. Caveolin-1 and cavin-1 play a required role in that their expression is necessary to stabilize cavin-3 levels. Caveolin-1 is an integral membrane protein that forms the core of the filamentous coat of caveolae. Cavin-1 has been proposed to act as an adaptor for the binding of other cavins. This published literature together with our data led to our proposal that cavin-3 bridges between the cavin-1/caveolin-1 complex and myosin-1/F-actin. Additional support for this conclusion comes from Figure 7, which as described in the response to critique 8, shows that cavin-3 is necessary for the ability of antibodies to myosin-1c to co-precipitate cavin-1 and caveolin-1. Literature in support of the model is now provided.